



# Estimating ocean tide loading displacements with GPS and GLONASS

Bogdan Matviichuk[1], Matt King[1], and Christopher Watson[1]

[1]School of Technology, Environments and Design, University of Tasmania, Hobart, 7001, Australia

**Correspondence:** Bogdan Matviichuk (bogdan.matviichuk@utas.edu.au)

**Abstract.** Ground displacements due to ocean tide loading have previously been successfully observed using GPS data, and such estimates for the principal lunar $M_2$ constituent have been used to infer the rheology and structure of the asthenosphere. The GPS orbital repeat period is close to several other major tidal constituents ($K_1$,$K_2$,$S_2$) thus GPS-estimates of ground displacement at these frequencies is subject to GPS systematic errors. We assess the addition of GLONASS to increase the accuracy and reliability over eight major ocean tide loading constituents: four semi-diurnal ($M_2$, $S_2$, $N_2$, $K_2$) and four diurnal constituents ($K_1$, $O_1$, $P_1$, $Q_1$). We revisit a previous GPS study, focusing on 21 sites in the UK and Western Europe, expanding it with an assessment of GLONASS and GPS+GLONASS estimates. In the region, both GPS and GLONASS data are abundant since 2010.0. We therefore focus on the period 2010.0-2014.0 which is considered long enough to reliably estimate the major constituents. Data were processed with a kinematic PPP strategy to produce site coordinate time series for each of 3 different modes: GPS, GLONASS and GPS+GLONASS. The GPS solution with ambiguities resolved was used as a baseline for performance assessment of the additional modes. GPS+GLONASS shows very close agreement with ambiguity resolved GPS for lunar constituents ($M_2$, $N_2$, $O_1$, $Q_1$) but substantial differences for solar-related constituents ($S_2$, $K_2$, $K_1$, $P_1$). While no single constellation mode performs best for all constituents and components, we propose to use a combination of constellation modes to recover tidal parameters: GPS+GLONASS for most constituents except for $K_2$ and $K_1$ where GLONASS (north and up) and GPS with ambiguities resolved (east), perform best.

## 1 Introduction

Earth's gravitational interactions with the Sun and the Moon generate solid-Earth and ocean tides. These tides produce periodic variations in both the gravity field and Earth's surface displacement. Additionally, the ocean tides produce a secondary deformational effect due to associated periodic water mass redistribution, known as Ocean Tide Loading (OTL) (e.g., Agnew, 2015; Jentzsch, 1997; Baker, 1984). OTL is observable in surface displacements (and their spatial gradients, i.e. tilt and strain) and gravity. Displacement and gravity attenuate approximately as the inverse of the distance from the point load while gradients have this relation but with distance squared (Baker, 1984). Thus, OTL displacement and gravity changes show greater sensitivity to regional solid Earth structure in comparison to tilt or strain observations Martens et al. (2016), making this an observation of interest for studying solid Earth rheology.





Global Navigation Satellite Systems (GNSS) are particularly convenient for measuring OTL displacements due to the widescale deployment of dense instrument arrays. Data from continuous GNSS stations have been shown to provide estimates of OTL with sub-millimetre precision using two main approaches as described by Penna et al. (2015): the harmonic parameter estimation approach – OTL displacement parameters are solved for within a static GNSS solution (e.g., Schenewerk et al., 2001; Allinson, 2004; King et al., 2005; Thomas et al., 2006; Yuan and Chao, 2012; Yuan et al., 2013); and the kinematic
approach – OTL constituents are estimated from high-rate kinematic GNSS-derived time series (e.g., Khan and Tscherning, 2001; King and Aoki, 2003; King, 2006; Penna et al., 2015; Martens et al., 2016; Wang et al., 2019). In this paper, we follow the kinematic approach.

To date, GNSS-derived OTL displacements have been estimated using predominantly the US Global Positioning System (GPS). GPS-derived measurements of Earth-surface displacement at tidal periods have been successfully used to observe
OTL displacement and validate ocean tide models (Urschl et al., 2005; King et al., 2005). The residual displacement between observed and predicted OTL has been related to deficiencies in ocean tide models, reference-frame inconsistencies, Earth model inaccuracies, the unmodelled constituents' dissipation effect and systematic errors in GPS (e.g., Thomas et al., 2006; Ito and Simons, 2011; Yuan et al., 2013; Bos et al., 2015).

Recent studies have made use of GPS-derived OTL to study dissipation or anelastic dispersion effects in the shallow as-
thenosphere at the $M_2$ frequency (e.g. Bos et al., 2015). This type of investigation has not been easily done previously due to various limiting factors. One key limitation was the limitations of the global seismic Preliminary Reference Earth Model (PREM) (Dziewonski and Anderson, 1981) that were demonstrated with GPS observations in the western United States (Ito and Simons, 2011; Yuan and Chao, 2012), western Europe (Bos et al., 2015), South America (Martens et al., 2016), Eastern China Sea region (Wang et al., 2019) and globally (Yuan et al., 2013). These limitations are associated partially with the in-
compatibility of the elastic parameters within the seismic (1 s period) and the tidal frequency bands and the anelasticity of the upper layers of the Earth, particularly the asthenosphere (Wang et al., 2019).

The use of OTL to probe the asthenosphere is increasingly possible because of accuracy advances of the global ocean tide models over recent time. Comparison of seven recent altimeter-constrained ocean-tide models to tide gauge and bottom pressure data shows agreement for eight major constituents of 0.9, 5.1 and 6.5 cm for pelagic, shelf and coastal conditions
respectively (Stammer et al., 2014). Bos et al. (2015) used this accuracy to infer that GPS-derived OTL displacement estimates were different to modeled displacements due to asthenospheric anelasticity. Lau et al. (2017) used results from the global study of Yuan et al. (2013) to constrain Earth's deep-mantle buoyancy.

Previous studies have highlighted an apparently large error in solar-related constituents estimated from GPS, in particular $K_2$ and $K_1$. This is in part due to their closeness to the GPS orbital ($K_2$) and constellation ($K_1$) repeat periods, which strongly
aliases with orbital errors. The closeness to the GPS constellation period may induce interference from other signals such as site multipath which will repeat with this same characteristic period (Schenewerk et al., 2001; Urschl et al., 2005; Thomas et al., 2006). Additionally, the $P_1$ constituent has a period close to that of 24 hours which is the timespan used for the IGS-standard orbit and clock products (ESA, but not CODE) (Griffiths and Ray, 2009), and hence may be contaminated by day-to-day discontinuities present in the products (Ito and Simons, 2011).





Urschl et al. (2005) proposed that the addition of GLONASS (GLObal NAvigation Satellite System), a GNSS developed and maintained by Russia (USSR before 1991), could improve the extraction of $K_2$ and $K_1$ constituents as the orbit period of the GLONASS satellites ( 11 h 15 min 44 sec) and constellation period ( 8 days) are well separated from major tidal frequencies. However, for many years GLONASS suffered from an unstable satellite constellation and very sparse network of continuous observing stations. This has been progressively addressed over the last decade to the point where many national networks now
include a high density of GLONASS (and other GNSS) receivers.

      In this publication, we expand the methodology described in Penna et al. (2015) with two more constellation configurations: stand-alone GLONASS and GPS+GLONASS; we also increase the number of assessed constituents to eight: $M_2$, $S_2$, $N_2$, $K_2$, $K_1$, $O_1$, $P_1$, $Q_1$. The dataset used in this study is completely overlayed by the original study dataset thus enabling a crosscheck between published GPS results. Afterwards, we intercompare the OTL estimates from GPS, GLONASS and a
combined GPS+GLONASS solutions acquired with different orbit and clock products in order to assess the constituent-specific sensitivities towards constellation/products configurations.

## 2   Dataset

Penna et al. (2015) validated GPS-derived estimates of OTL for geophysical interpretation while Bos et al. (2015) followed the previous study to analyse OTL displacements in western Europe. As previously stated, to quality control our analysis
we decided to follow the validation approach and study region of Penna et al. (2015), but extended it to include GLONASS and GPS+GLONASS solutions. Aside from the addition of GLONASS data, an important difference to the original studies is the shift in time period from 2007.0–2013.0 to 2010.0–2014.0. This shift provides sufficient GLONASS data following the upgrade of many receivers to track GNSS from 2009. Despite this covering a shorter time span, the length of continuous observation within each site (minimum availability of 0.95 through dataset) exceeds the recommended  1000 days of continuous
observations (4 years with 0.7 availability) (Penna et al., 2015). Additionally, selected time period is fully covered by a complete set of reprocessed orbit and clock products.

      The sites used in our study are shown in Figure 1, with a focus on south-west England where a large $M_2$ OTL signal is present. Of the 21 stations, 14 stations are in south-west England and the southern part of Wales with an additional seven stations in Europe, all equipped with GPS+GLONASS receivers. Our station set is somewhat different to that used by Penna
et al. (2015) due to the lack of GLONASS-capable receivers, replacement with sites a few metres away, or the addition of new sites which were deployed just before 2010.0.

      The chosen sites experience a range of $M_2$ up OTL amplitudes ranging from > 30 mm (ANLX, APPL, BRST, CAMO, PADT, PRAE), 15-30 mm (CARI, EXMO, LOFT, PBIL, SWAS, TAUT) and < 15 mm (BRAE, CHIO, LERI, POOL, SANO, WEAR, ZIM2). Because of the wide range of experienced OTL within the dataset, a detailed sensitivity assessment of con-
stituent/constellation configurations pairs became possible.



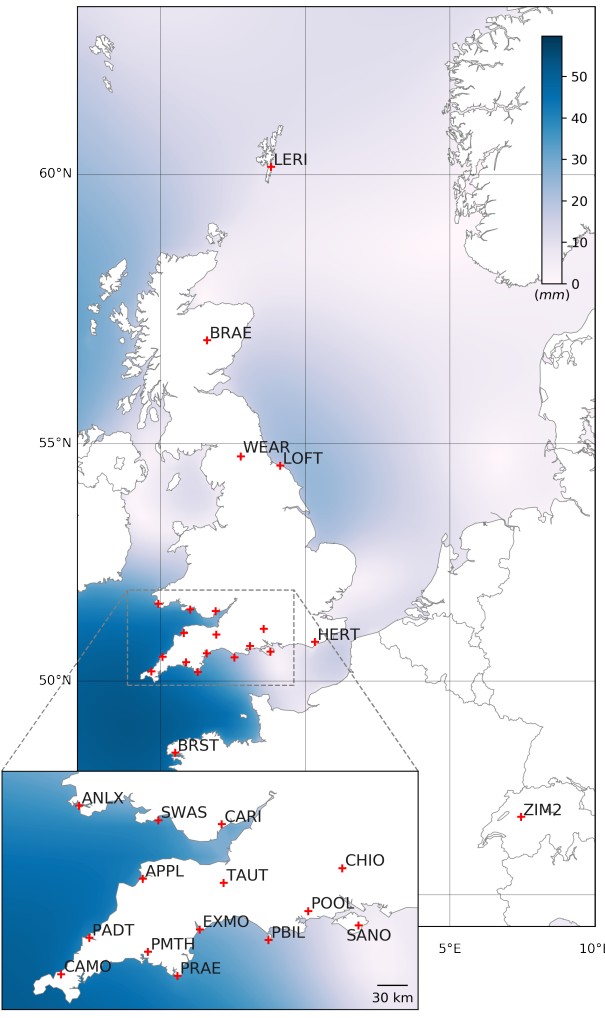

**Figure 1.** Map of the study area with GNSS site codes and $M_2$ up displacement amplitude in the background (TPXO.7.2 ocean tide model and spherically symmetric earth with PREM structure).

## 3 GNSS data processing strategy

The processing strategy was largely based on the GPS-only kinematic Precise Point Positioning (PPP) approach (Zumberge et al., 1997) as per Penna et al. (2015), but with modifications in terms of the software and to permit the inclusion of GLONASS data. We address PPP in three different modes here: GPS, GLONASS and combined GPS+GLONASS. In particular, we use

NASA JPL's GipsyX (v. 1.3), which is a substantial rewrite of the now legacy GIPSY-OASIS code to allow for, amongst other things, multi-GNSS analysis. Penna et al. (2015) used GIPSY-OASIS v6.1.2. We adopted a PPP solution approach and estimated station positions every 5 minutes with a random walk model introducing estimated optimum between-epoch



constraints on coordinate evolution. We used the VMF1 gridded troposphere mapping function, based on the European Centre for Medium-Range Weather Forecasts (ECMWF) numerical weather model (Boehm et al., 2006). Additionally, ECMWF

values for the hydrostatic zenith delay and wet zenith delay were used as a priori values for stochastic estimation of the wet zenith delay as a random walk process with optimum process noise values (Sect. 4) and tropospheric gradients were estimated as a random walk process (Bar-Sever et al., 1998), with process noise at 0.005 mm/sqrt(s). An elevation cut-off angle of seven degrees was applied together with observation weights a function of the square-root of the sine of the satellite elevation angle.

Earth body-tide (EBT) and pole tides were modelled according to IERS Conventions (Petit and Luzum, 2010). The OTL

displacement within processing run was modelled with the FES2004 tidal atlas (Lyard et al., 2006) and elastic Green's functions based on the Gutenberg-Bullen Earth model (Farrell, 1972) (FES2004_GBe), with centre-of-mass correction applied depending on the adopted orbit products. The OTL values were generated in BLQ format for 11 principal constituents ($M_2$, $S_2$, $N_2$, $K_2$, $K_1$, $O_1$, $P_1$, $Q_1$, $Mf$, $Mm$ and $Ssa$) using free ocean tide loading provider (http://holt.oso.chalmers.se/loading).

PPP requires pre-computed precise satellite orbit and clock products for each constellation processed which should be solved

for simultaneously within a single products solution. Unfortunately, JPL's native clock and orbit products are not yet available for non-GPS constellations hence we adopted products from two International GNSS Service (IGS) (Johnston et al., 2017) Analysis Centres (ACs): the European Space Agency (ESA) and Centre for Orbit Determination in Europe (CODE). The ESA combined GPS+GLONASS Products from the IGS second reprocessing campaign (repro2) were used (Griffiths, 2019) while CODE's more recent REPRO_2015 campaign (Susnik et al., 2016) had to be used as CODE's repro2 are lacking separate 5

min clock corrections.

All three products consist of satellite orbits and satellite clock corrections, sampled at 15 and 5 minutes respectively, that were held fixed during our processing. The benefit of using JPL's native products, even though solely GPS, is the ability to do PPP processing with integer ambiguity resolution (AR). PPP AR in GIPSY-OASIS/GipsyX software can be performed by using wide lane and phase bias tables which are part of JPL's native products (Bertiger et al., 2010). To provide comparison

with previous studies, GPS was processed with JPL's native orbit and clock products from the repro2 campaign (JPL's internal name is repro2.1) with AR.

The CODE and ESA clock and orbit products were generated in different ways. CODE's REPRO_2015 orbit positions were computed using a 3-day data arc, while ESA used a 24-h data arc (Griffiths, 2019). Both ACs provided orbits in a terrestrial reference frame, namely IGS08 and IGb08, respectively, that are corrected for the centre of mass motion associated with OTL

(FES2004 centre of mass correction) and are in the CE frame, following Blewitt (2003). Alternatively, JPL products were generated from a 30-h data arc, and were computed with stations in a near-instantaneous frame realisation hence the orbits are in the CM frame (we note that the JPL products distributed by the IGS are, by contrast, in CE). Considering the above, the modelled OTL values for JPL's native products solutions were corrected for the effect of geocentre motion while ESA/CODE products do not require this correction (Kouba, 2009).

It has been suggested that orbit arc length for a given product could potentially impact the estimated OTL displacements. In particular, Ito and Simons (2011) suggest that a 24-h data arc length (as per ESA products) may affect the $P_1$ constituent due to similarity of the periods. This is in addition to day-boundary edge effects given analysis of data in 24-h batches. We mitigate





these effects to some extent by processing the ground stations in 30-h batches (allowing 3-h either side of the nominal 24-h day boundary) and merging the IGS-standard 24-h orbits/clocks into 30-h where necessary.

We post-processed the estimated coordinate time series as per Penna et al. (2015): the resulting 5-min sampled solutions were clipped to the respective 24-h window and merged together. The raw 4-yr timeseries were filtered, converted to a local east-north-up coordinate frame, detrended and resampled to 30-min sampling rate via a simple 7-point window average (7 samples -> 1 sample). 30-min averaging reduces high frequency noise (unrelated to OTL) as well as the computational burden of further harmonic analysis.

Finally, OTL displacements modelled in GipsyX were added back using HARDISP (Petit and Luzum, 2010). HARDISP uses spline interpolation of the tidal admittance of 11 major constituents to infer values of 342 tidal constituents and generate a time series of tidal displacements. This approach almost eliminates the effect of companion constituents (Foreman and Henry, 1989) as they are modelled during processing stage, even considering errors present in companion constituent displacement as tide model errors become neglectable for constituents this small. Thus, analysed harmonic displacement parameters represent

true displacement plus indiscernible companion constituent error, far below measurement error.

The harmonic analysis of the reconstructed OTL signal was performed using Eterna software v.3.30 (Wenzel, 1996) with a high-pass filter (30-min sampling), resulting in amplitudes and local tidal potential phase lags negative which are suitable for the solid Earth tide studies. OTL phase-lag, however, is defined with respect to the Greenwich meridian and phase lags positive.

The latter, however, is not important for the purposes of this publication as we ignore the long-period tides in the analysis

but used them for internal Eterna testing; same goes for diurnal tides phase rotation as all sites' latitudes are positive here. Transforming to Greenwich-relative lags was done using the procedure according to Boy et al. (2003) and Bos (2000).

We then computed the vector difference between the observed OTL vector and that predicted by the model, following the notation of Yuan et al. (2013):

$$Z_{res} = Z_{obs} - Z_{th} \tag{1}$$

In Eq. 1 we assume body tide errors to be negligible, thus $Z_{obs}$ is simply an observed OTL and $Z_{th}$ is a theoretical OTL. Residual OTL, $Z_{res}$, is the difference between OTL model predictions and observational errors. The residual OTL presented in this paper is, if not otherwise specified, relative to the theoretical OTL values computed using the FES2014b ocean tide atlas, a successor of FES2012 used in (Bos et al., 2015), and a Green's function based on the STW105 Earth model additionally corrected for dissipation at the $M_2$ frequency which we call STW105d (FES2014b_STW105d).

**4   Process noise optimization**

Process noise settings within GipsyX need to be chosen to ensure optimal separation of site displacement, tropospheric zenith delays, noise etc. For example, tight coordinate process noise value, even the default value of 0.57 mm/sqrt(s), tend to clip OTL amplitudes, especially in costal sites. Penna et al. (2015) developed a method of tuning process noise values for GPS PPP,





which we expanded to accommodate the additional major diurnal/semidiurnal constituents considered here, as well as the use
of GPS and GLONASS data.

To do this, we used the CAMO site, the successor of CAMB used by Penna et al. (2015) and tested a range of coordinate and
Zenith Wet Delay (ZWD) process noise settings exactly as described by Penna et al. (2015). These tests focus on a range of
metrics, namely the standard deviation of the height time series (shown as "Ht std/3", as divided by 3), the standard deviation
of kinematic ZWD normalized by ZWD values from a static solution ("ZWDstatic"), root mean square of the carrier phase
residuals ("RMSres"), magnitude of $M_2$ residual OTL, $\|Z_{res}\|$,computed by differencing observed OTL with FES2004_GBe
theoretical values ("$M_2$") and $\|Z_{res}\|$ of a synthetic 13.96 h signal and its controlled, known input (designated "synth err "").
The "synth err" $\|Z_{res}\|$ was estimated from solutions with harmonic signal introduced into sites' nominal location (2, 4 and 6
mm into east, north and up components respectively). The extracted amplitudes were then compared with the known signal to
measure the level of propagation into other components.
For each of the major constituents, both diurnal and semi-diurnal, we found that 3.2 mm/sqrt(s) for coordinate process noise
and 0.1 mm/sqrt(s) for tropospheric zenith delay process noise were optimal for our solutions, the same values as identified by
Penna et al. (2015). Figure 2 shows the results of the tests, with the left panel showing the result of varying coordinate process
noise while ZWD process noise was held fixed (0.1 mm/sqrt(s), a default value) and the right panel the result of varying the
ZWD process noise with coordinate process noise equal to the optimum value of 3.2 mm/sqrt(s). The only difference in our
results to those of Penna et al. (2015) were for the "synth err" test, where our results are inverted (but without changing the
magnitude); the reason for this is discussed in detail in the supplementary material.

## 5 Results and Discussion

### 5.1 Effect of using GLONASS

Given the known accuracy of the ocean tide models in this region, and small effects of errors in solid Earth models, our as-
sumption is that as $\|Z_{res}\|$ approaches zero the estimates increase in accuracy, as in Bos et al. (2015). Figure 3 shows GPS,
GLONASS and GPS+GLONASS $\|Z_{res}\|$ estimates for east, north and up coordinate components. The combined GPS+GLONASS
solutions perform either at the same level as GPS AR ($M_2$, $O_1$, $Q_1$) or better ($N_2$, $P_1$) for the up component. $\|Z_{res}\|$ val-
ues are smaller and more consistent for the east ($M_2$, $N_2$, $O_1$) and north ($M_2$, $N_2$, $P_1$) components respectively. Also, the
GPS+GLONASS solution does not have $\|Z_{res}\|$ biases in the east and north components as is noticeable for the GPS AR
solution (particularly for $O_1$ in east, and $P_1$ in north, respectively). By $\|Z_{res}\|$ bias we mean a noticeable gap between zero and
the distribution's lower bound (25th percentile - 1.5*interquartile range), which is present at all sites no matter how far inland.

Considering the problematic GPS $K_2$ and $K_1$ constituents, the GPS AR can reasonably reliably, in comparison to other
types of solutions, extract $\|Z_{res}\|$ in the east component (Figure 3, lower left panel) which is smaller than that of GLONASS
and GPS+GLONASS using ESA or CODE products. However, smallest $\|Z_{res}\|$ extraction in the up and north components is
possible only using GLONASS constellation solely.







**Figure 2.** The effect of varying coordinate process noise (left) and ZWD process noise (right) at test site CAMO for the up component (2010.0 – 2014.0), performed with ESA repro2 products. We expanded the tests of Penna et al. (2015) to include $\|Z_{res}\|$ of seven additional major constituents ($S_2$, $N_2$, $K_2$, $K_1$, $O_1$, $P_1$ and $Q_1$), named accordingly, and different constellations. The different constellations' configurations: GPS, GLONASS and GPS+GLONASS are presented as solid, dashed and dotted lines respectively. The colours pertain to the different metrics as described in the text and legend (note the same scheme is used as per Penna et al. (2015).

Our results suggest that no single solution provides consistently better constituent estimates across all coordinate components. We suggest that optimum results are obtained using GPS+GLONASS for $M_2$, $N_2$, $O_1$, $P_1$ and $Q_1$, and GLONASS for $K_2$ and $K_1$, noting that GPS AR performs better for the $K_2$ and $K_1$ in the east component.

We now explore the sensitivity of our solutions to different products and analysis choices.





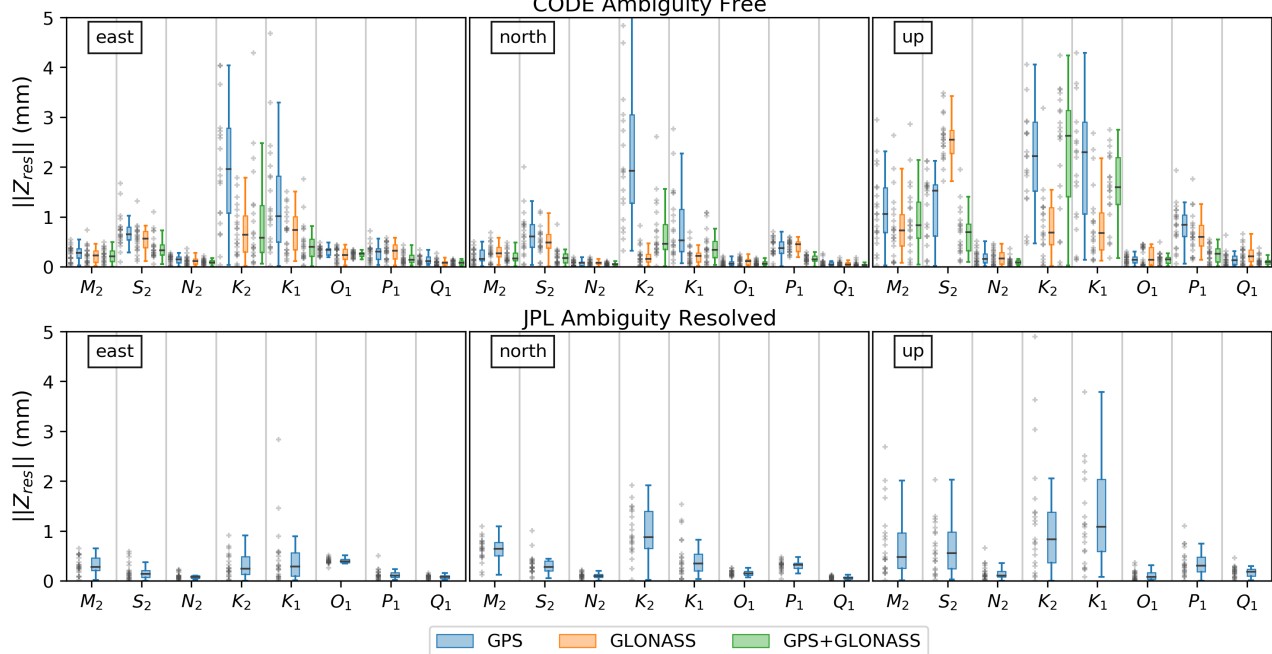

**Figure 3.** $\|Z_{res}\|$ per tidal constituent for east, north and up components (left, middle and right, respectively) relative to FES2014b_STW105d OTL values with CMC correction for JPL solutions. Grey crosses to the left of each boxplot represent sites' $\|Z_{res}\|$ values and are offset horizontally for clarity while the horizontal line over each boxplot is a median of each constituent's $\|Z_{res}\|$. Top: $\|Z_{res}\|$ for GPS, GLONASS and GPS+GLONASS PPP solutions (blue, orange and green, respectively) computed using CODE products. Bottom : $\|Z_{res}\|$ of the GPS AR solution computed with JPL native products.

## 5.2 Satellite orbit and clock products sensitivity tests

We assessed whether the solutions were sensitive to changes in satellite-elevation cutoff angle. Three additional cutoff angle scenarios were tested: 10°, 15° and 20° (in addition to the default 7° cutoff angle). Different elevation cutoffs will significantly alter the observation geometry as well as modulate the expression of signal multipath into solutions.

Figure 4 (top) shows the magnitude of vector distance, $\|\Delta Z_{res}\|$, between estimated $Z_{res}$ values estimated from the 7° and 20° solutions and CODE products in both cases (upper subplot). $S_2$, $K_2$, $K_1$ and $P_1$ constituents in the up coordinate component show large mean vector distances in both GPS (0.56, 2.29, 2.88, 0.54 mm, respectively) and GLONASS (0.82, 0.64, 1.01, 0.58 mm, respectively) with the rest of constituents showing differences of less than 0.5 mm. GPS+GLONASS solution, up component, minimizes the $\|\Delta Z_{res}\|$ of $S_2$ and $P_1$ (0.31, 0.23 mm, respectively) and shows an additional decrease in $\|\Delta Z_{res}\|$ for $M_2$, $S_2$,$N_2$, $O_1$, $Q_1$ in the up component. For the horizontal coordinate components, GPS+GLONASS minimizes $\|\Delta Z_{res}\|$ for all constituents it increases the stability of all eight major constituents (including $K_1$ and $K_2$).

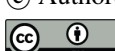



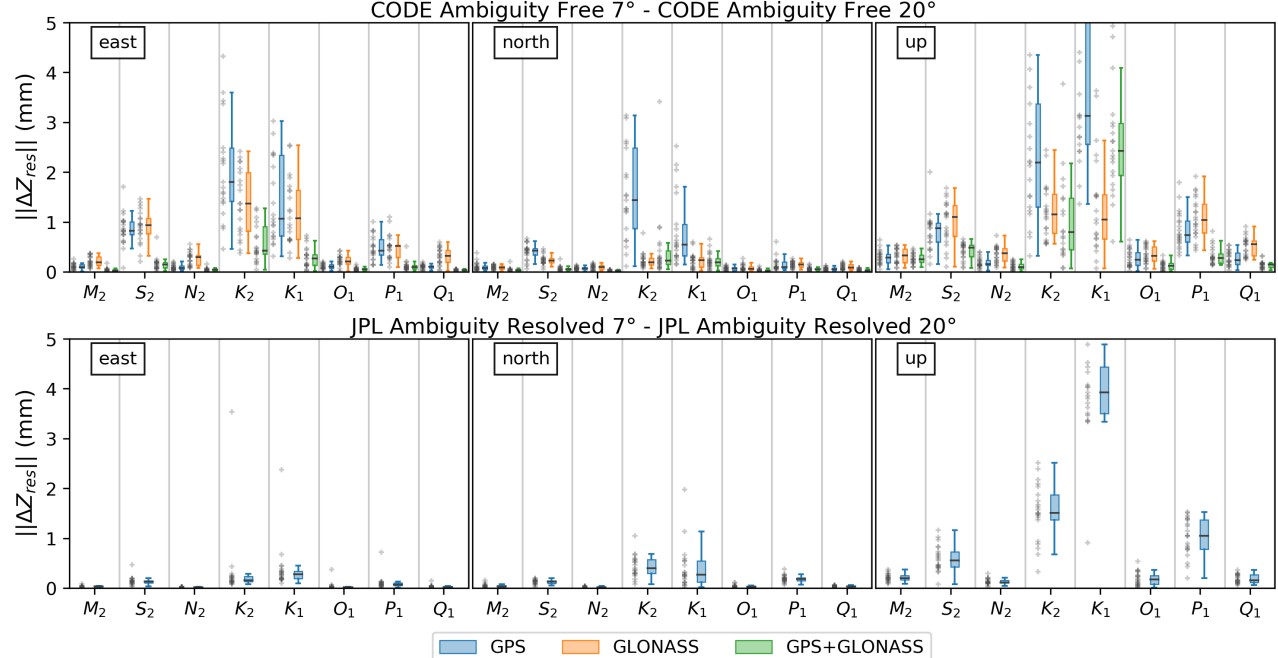

**Figure 4.** Magnitude of vector distance between estimated $Z_{res}$ values computed with 7° and 20° elevation cutoff angles, $\|\Delta Z_{res}\|$, within the same set of orbits and clocks (top: CODE; bottom: JPL AR ) for east, north and up coordinate components (left, middle and right, respectively). Grey crosses are as per figure 3. Smaller residuals of CODE's GPS+GLONASS (top) is a result of better stability of OTL estimated with combined constellations towards various cutoff angles (except $K_1$ up and $K_2$ up). JPL's GPS AR also shows great stability with exception of $K_2$ up and $K_1$ up. $\|\Delta Z_{res}\|$ for GPS, GLONASS and GPS+GLONASS PPP solutions in blue, orange and green, respectively.

The same comparison was done for GPS AR 7° and 20° solutions (JPL native products) and shows largely improved stability in comparison to CODE's GPS solution which has effectively the same performance as with JPL's GPS only products (Figure 4, bottom). However, $K_2$ up and $K_1$ up show substantial differences between solutions: $K_2$ due to a lot tighter distribution of $\|Z_{res}\|$ in the 20° solution, possibly due to removal of multipath, and $K_1$ due to increased dispersion and median of $\|Z_{res}\|$ at
increased cutoff angle.

Following Yuan et al. (2013), we assessed the possible influence of inconsistencies in precomputed orbits/clocks on estimated OTL displacements. This was done by computing $\|\Delta Z_{res}\|$ between pairs of solutions with common constellation configurations: GPS (no AR here) solutions computed using ESA, CODE and JPL products; GLONASS/GPS+GLONASS solutions using ESA and CODE products. The importance of this assessment is directly related to the main principle of PPP
technique: a priori knowledge of satellite-related parameters that are held fixed when estimating station coordinate time series. Given this approach, all satellite-related systematic errors will have at least partial expression into station-specific parameters (Yuan et al., 2013).





**Figure 5.** OTL vector differences between CODE, ESA and JPL solutions (ambiguity free). Top: GPS, GLONASS and combined GPS+GLONASS differences between CODE and ESA solutions; Middle: GPS difference between CODE and JPL solutions (ambiguity free); Bottom: GPS difference between ESA and JPL solutions (ambiguity free). Note the vertical scale of 2 mm. Grey crosses are as per Figure 3.

Figure 5 (top) shows the distribution of $\|\Delta Z_{res}\|$ between solutions computed with ESA and CODE products for all three constellation modes: GPS, GLONASS and GPS+GLONASS. The main differences are related to the $S_2$, $K_2$, $K_1$ and $P_1$

constituents. The maximum $\|\Delta Z_{res}\|$ between the observed OTL for the rest of the constituents is less than 0.3 mm.

Compared with GPS JPL, both CODE and ESA solutions (Figure 5, middle and bottom, respectively) show $\|\Delta Z_{res}\|$ up to 0.5 mm in the horizontal components with respect to JPL solutions, which is also true for ESA in the up component with exception for $K_2$ and $K_1$. CODE shows similar behaviour to ESA, however, significant divergence from JPL (Figure 5, middle) is also observed for $S_2$ with even higher $\|\Delta Z_{res}\|$ for $K_2$ and $K_1$ in the up and the east.





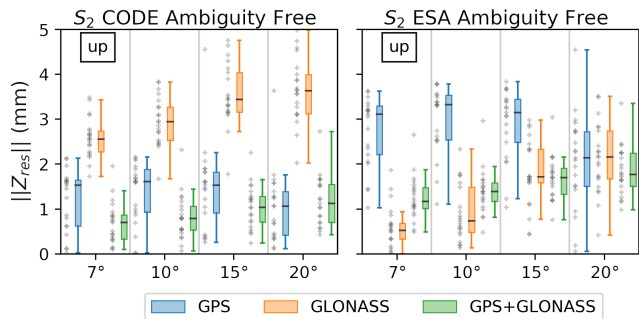

**Figure 6.** GPS, GLONASS and GPS+GLONASS $\|Z_{res}\|$ for the $S_2$ constituent in the up component as a function of elevation cutoff angle, computed with ESA (left) and CODE products (right). Note the inverse behaviour of GPS and GLONASS biases and the linear dependence of the GLONASS biases. Grey crosses are as per Fig. 3.

### 5.3  $S_2$ constituent

Focusing on $S_2$, the GPS up residual shows 1 mm residual bias between solutions using CODE and ESA products (compare blue records between left and right panels, Figure 6). The GPS $\|Z_{res}\|$ bias is maintained for solutions with a range of elevation cutoff angles (7°, 10°, 15° and 20°). GLONASS solutions (orange), however, show no $\|Z_{res}\|$ bias for ESA and 2 mm bias for CODE, both with 7 ° elevation angle. GLONASS bias values in both cases increase with elevation cutoff angle up to 15°. This GLONASS dependency with elevation cutoff is present to a lesser degree in both east and north components and is the same with ESA and CODE products.

GPS $\|Z_{res}\|$ estimates show inverse behaviour in terms of $\|Z_{res}\|$ bias between ESA and CODE solutions in the up component (blue, Figure 6) and an additional linear dependency of increasing median $\|Z_{res}\|$ as a function of increasing elevation cutoff in the north component. Both ESA and CODE GPS+GLONASS $S_2$ results (green, Figure 6) show a blend of the two patterns observed with GPS and GLONASS solutions. GPS+GLONASS $S_2$ shows less sensitivity to the cutoff angle change than the GLONASS solutions with both products. However, the CODE GPS+GLONASS $S_2$ solution consistently produces smaller $\|Z_{res}\|$ biases which may be related to the absence of $\|Z_{res}\|$ bias in the GPS solution and its higher weight in the solution due to higher number of simultaneously visible SVs. The effect was also studied with JPL products and AR: GPS and GPS AR; see Sect. 5.7.

The substantial difference in $S_2$ between ESA and CODE (Figure 6) suggests important differences in raw GNSS data analysis approaches within respective Analysis Centres. One possible explanation is associated with the treatment of S1 and $S_2$ atmospheric tides which were corrected for at the observation level in CODE products (no correction in ESA). However, the total value of the vector difference cannot be explained by unmodelled atmospheric tides alone. Additionally, the inverse behaviour of GPS and GLONASS between ESA and CODE solutions (orange, Figure 6) cannot be explained with a single correction applied to both constellations. We expect that the differences in each solution are a function of satellite orbit modelling, although the issue is not clear and needs further investigation.



### 5.4  $K_2$ and $K_1$ constituents

Like 7° CODE and ESA solutions, $\|\Delta Z_{res}\|$ can be minimized if using GLONASS for the extraction of $K_1$ and $K_2$ constituents and GPS+GLONASS for the remainder of the constituents. In this case, $\|\Delta Z_{res}\|$ will stay below 0.25 mm for north
components and below 0.5 mm for the east and the up components.

GLONASS $K_2$ and $K_1$ estimates in the north have the tightest distribution of the $\|Z_{res}\|$, are most stable for the range of elevation cutoff angles and products. For the east component, CODE products GLONASS struggles to compete with GPS+GLONASS in terms of minimised distribution of $\|Z_{res}\|$ ($K_1$) and elevation cutoff stability ($K_2$ and $K_1$). The GLONASS $K_1$ east is not true for the ESA products solutions which overperform the respective GPS+GLONASS in terms of $\|Z_{res}\|$ distri-
bution consistency and median (see supplementary Figure $S_2$) and this explains the minor difference between ESA and CODE GLONASS solutions (Figure 5, top left). Elevation cutoff stability of ESA $K_2$ and $K_1$ in the east component is exactly as with CODE – best with GPS+GLONASS.

The up component of $K_2$ and $K_1$ is the most problematic, showing high $\|Z_{res}\|$ values with all constellation modes. GLONASS OTL values from both ESA and CODE solutions have tightest distributions and smallest $\|Z_{res}\|$ median values
overperforming JPL GPS AR. Note that GPS+GLONASS $K_2$ up has marginally smaller median $\|\Delta Z_{res}\|$ in the elevation cutoff test, possibly due to higher amount of SVs, however, $\|Z_{res}\|$ values recovered are too high with major differences between CODE and ESA products' values.

While we cannot certainly select a single constellation configuration optimal for all components of $K_2$ and $K_1$, we can conclude that based on our analysis, GLONASS performs best in the $K_2$ and $K_1$ north and up components while east component
might show better results with GPS+GLONASS ($K_1$, CODE) but, due to better consistency between products, GLONASS values are still preferred. The only exception to the east component conclusions is GPS AR (see Sect. 5.7).

### 5.5  $P_1$ constituent

The high $\|\Delta Z_{res}\|$ in GLONASS $P_1$ constituent between CODE and ESA solutions over all coordinate components (orange, Figure 5 top) is unexpected as ESA and CODE $\|Z_{res}\|$ boxplots show similar distributions of values (see Figure $S_2$ in sup-
plementary material for the equivalent ESA boxplots). This suggests a symmetrical deviation from the modelled values that produces a high $\|\Delta Z_{res}\|$. In all cases, however, GPS+GLONASS is preferred for $P_1$ estimation.

### 5.6  Effect of different orbit and clock products on noise and uncertainty

Changing orbit and clock products also changes the time series noise characteristics and hence influences the uncertainties of the estimated constituents (estimated separately by Eterna for amplitude, Figure 7 and phase, Figure 8). Amplitude uncer-
tainties are expressed here as an average across all constituents as they do not differ much between analysed constituents. ETERNA assumes a white noise model in its analysis. We conclude that GLONASS solutions produce the highest amplitude uncertainties for east (0.15 mm CODE, 0.14 mm ESA) and up components (0.22 mm CODE, 0.27mm ESA) while showing the same uncertainty as GPS for the north (0.07 mm, both CODE and ESA). GLONASS amplitude uncertainties from solutions





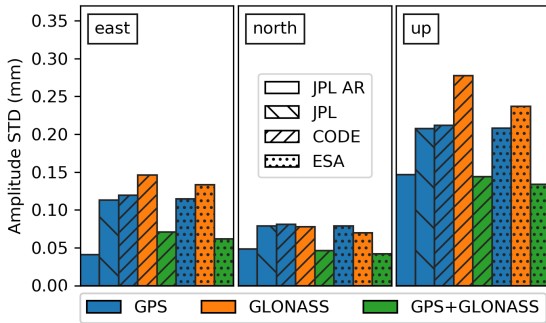

**Figure 7.** Average uncertainties (1-sigma) for OTL amplitudes computed across eight OTL constituents per products (stipple) and processing modes/constellation (colour): GLONASS (orange) and GPS (blue) modes show higher uncertainties STDs, while GPS-only AR and combined GPS+GLONASS (green) show minimum uncertainties with exception for east.

using CODE products tend to be marginally higher than those of ESA products. The amplitude uncertainties for combined
GPS+GLONASS solutions are equal to those of JPL with ambiguities fixed (GPS AR), although the JPL GPS AR solution has slightly smaller uncertainty in the east component (smaller by 0.02 mm).

Considering the uncertainties of phase values, these are unsurprisingly dependent on the constituent's amplitude. JPL native products show a significant advantage in this case when compared to ESA and CODE due to differences in frames: CM (JPL) and CE (ESA and CODE). This results in up to an order of magnitude increase in phase uncertainties for "weaker" diurnal
constituents in the region: $N_2$, $O_1$, $P_1$, $Q_1$ (Figure 8).

In general, this frame effect is directly related to centre of mass correction (CMC) specific to the constituent's CMC vector in comparison to the total theoretical OTL vector. If applying a CMC correction to the constituent increases its amplitude, phase STD values will decrease in CM-frame solution. This is critically important for the constituents with amplitudes below 0.5 mm, as phase uncertainty increases significantly below this threshold. The most significant exception in our dataset is $P_1$
in the up component which has a much larger amplitude in CE frame (Figure 8, right in top and bottom).

Converting CE products to CM (Figure 8, bottom) was done to demonstrate that the changes in phase uncertainty are indeed introduced by the smaller amplitudes in the CE frame. While this holds true, it is obvious that not only does the P1 up phase uncertainty increase, as was expected based on comparison with JPL solutions. GLONASS $K_1$ up phase uncertainties show almost an order of magnitude increase in CM frame while having unexpectedly small values in CE. This is a direct cause of
GLONASS solution having larger $K_1$ up amplitudes in CE and smaller in CM with both CODE and ESA.

## 5.7 Impact of Ambiguity Resolution on GPS

The multi-GNSS products used here do not allow integer AR with PPP and this is an active area of research and development within the IGS. However, assessing the impact of AR on GPS-only solutions provides some insight towards the future benefit of AR on GLONASS and GPS+GLONASS solutions once such products become available. We compared OTL residuals from





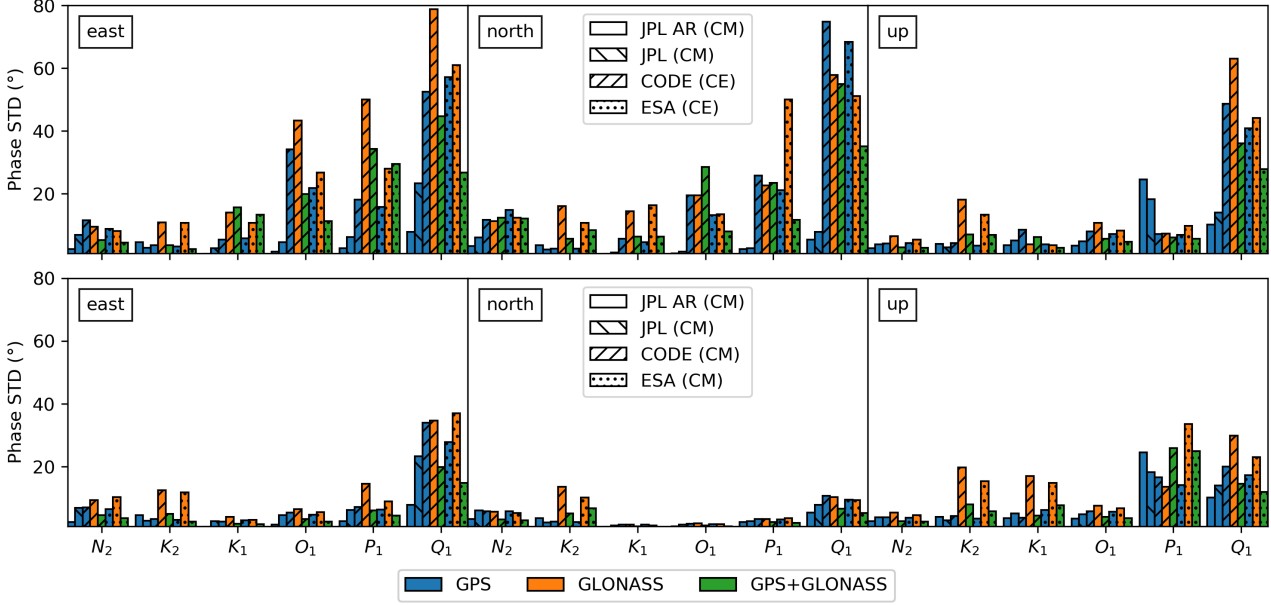

**Figure 8.** Average phase STD (uncertainty) per constituent for different products. ESA and CODE products were in CE frame by default (top) and converted to CM (bottom) while JPL are in CM in both. $M_2$ and $S_2$ phase STDs are not shown here as values are too small to be seen with the scale specified.

GPS and GPS AR using JPL native products that contain wide lane and phase bias tables (WLPB files) required for integer AR with PPP.

Figure 9 shows the effect on estimated constituents from enabling AR in a standard solution with 7° cutoff. Here we observe decreased $\|Z_{res}\|$ over all coordinate components compared with the estimates from a non-AR solution. This is most visible in the $K_2$ and $K_1$ constituents and in the elimination of the $S_2 \|Z_{res}\|$ bias and with smaller improvements in $M_2$ and $P_1$.

Importantly, Figure 9 shows that enabling AR eliminates $\|Z_{res}\|$ bias in GPS and aligns the residual vectors with ESA/CODE GPS+GLONASS (Figure 3). Thus, the $S_2 \|Z_{res}\|$ bias was once again assessed with various elevation cutoff angles solutions. JPL GPS solutions, in the up component (Figure 10, left), show $S_2 \|Z_{res}\|$ bias to be constant between cutoff angles at about 1 mm with median $\|Z_{res}\|$ fluctuating around 3 mm. Similar behaviour was previously observed with solutions using ESA products.

Enabling integer ambiguity resolution (GPS AR) removes the $S_2 \|Z_{res}\|$ bias completely over all tested elevation cutoff angles while introducing a slight increase of the median residual magnitude in the up component with increasing elevation cutoff. Based on this observation, we expect that utilising ambiguity resolution within PPP might help in solving, or at least minimising, the $S_2 \|Z_{res}\|$ present in ESA GPS and CODE GLONASS solutions. Eliminating biases in GPS and GLONASS separately should increase the stability and consistency of GPS+GLONASS $S_2 \|Z_{res}\|$.

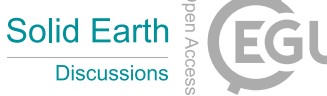

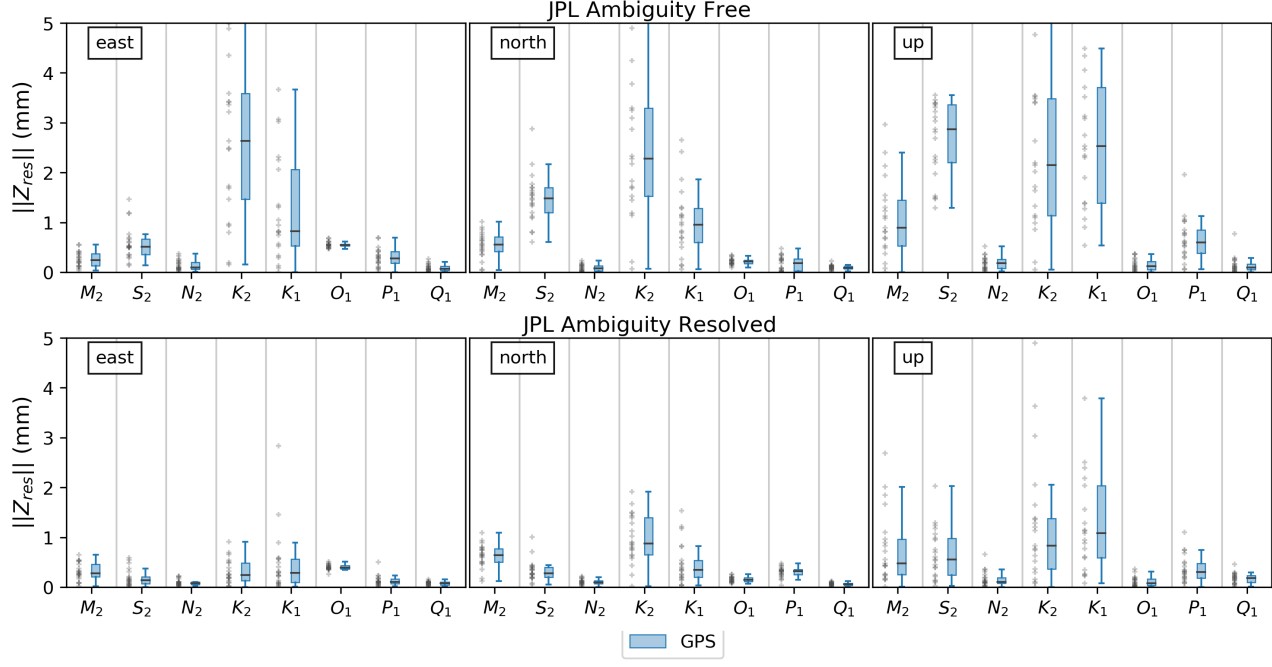

**Figure 9.** Comparison of residual constituents' estimates from GPS (top) and GPS AR (bottom) JPL native solutions. Grey crosses are as per Figure 3. As seen, most of constituents' $\|Z_{res}\|$ distributions became tighter and medians smaller while $S_2$ $\|Z_{res}\|$ bias got removed with enabling of AR.

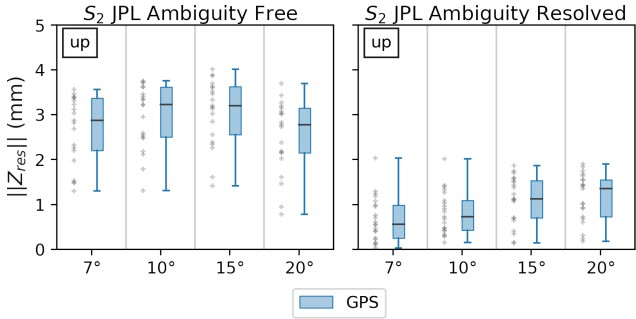

**Figure 10.** GPS $S_2$ up constituent's $\|Z_{res}\|$ change with elevation cutoff angle computed with JPL products floating AR (left) and integer AR (right). Grey crosses are as per Figure 3. As seen, AR completely removes the bias and slightly improves overall consistency between stations.

## 5.8 Impact of timeseries length

Yuan et al. (2013) used filter based harmonic parameter estimation approach and demonstrated the dependence of Kalman filter convergence and timeseries length for each of the eight major constituents. Yuan et al. (2013) concluded that after 1000





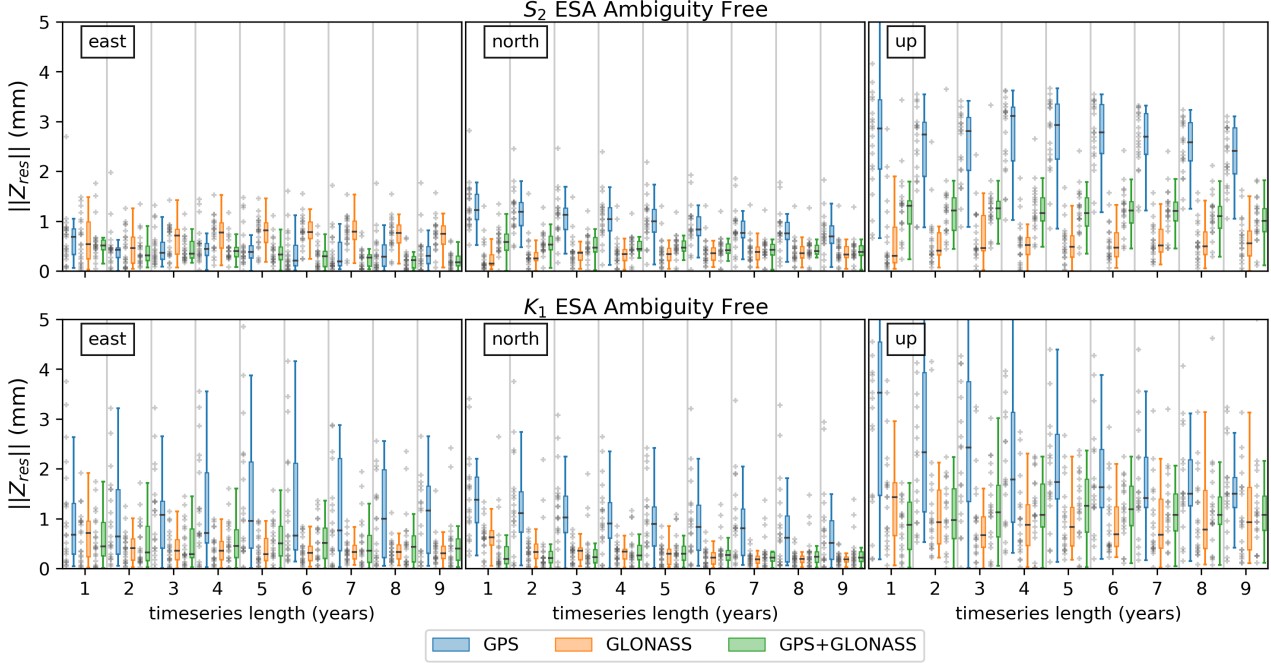

**Figure 11.** Dependency of estimated $\|Z_{res}\|$ and timeseries' length in years for two solar related constituents: $S_2$ (top), $K_1$ (bottom). GPS, GLONASS and GPS+GLONASS PPP solutions in blue, orange and green, respectively using ESA products. Grey crosses are as per Figure 3. Note that 1 to 4 years of timespan use ESA repro2 while the rest uses a combination of ESA repro2 and ESA operational products.

daily solutions convergence (minimized $\|Z_{res}\|$) was reached for lunar-only constituents ($M_2$, $N_2$, $O_1$, $Q_1$) while reporting solar-related constituents ($S_2$, $K_2$, $K_1$, $P_1$) were not fully converged even after 3000 daily solutions.

Here, we assessed how $\|Z_{res}\|$ of each of 8 major constituents vary as a function of the time series length with kinematic estimation approach. The duration of the series varied by integral years, and to enable a complete analysis, we expanded the candidate solutions to 2019.0 and processed additional data with operational products: JPL repro3.0, ESA operational, CODE MGEX (CODE operational lack GLONASS clock corrections). Importantly, the results shown in Figure 11 are therefore a composition of reprocessed products and operational products (years 5 to 9); the design of the reprocessed solutions should

closely match the operational products.

    Figure 11 shows that even three-years of data appears to be sufficient to get reliable $\|Z_{res}\|$ values for GLONASS for $K_1$ and GLONASS or GPS+GLONASS for $S_2$ (see Supplementary Figure S3 ).

    Changing satellite orbit and clock products may produce substantial differences in results. Thus, we performed a comparison of ESA repro2 solutions (2010.0-2014.0) with the ESA operational product (2014.0-2019.0) which showed no significant

changes in terms of $\|\Delta Z_{res}\|$ for $M_2$, $N_2$, $O_1$, $P_1$ and $Q_1$ estimated (Figure 12). GLONASS, however, shows significant difference between two datasets for $K_1$ and $K_2$, up specifically, which might be related to the changes in GLONASS products





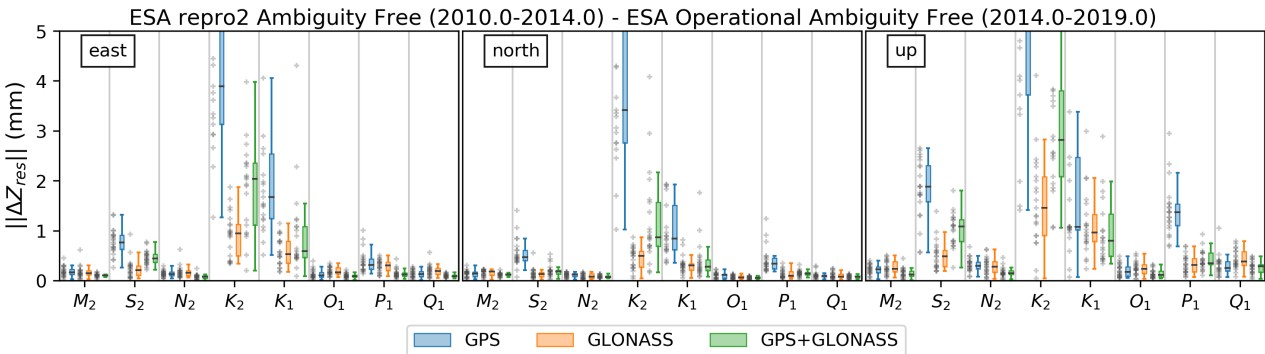

**Figure 12.** OTL vector differences between ESA repro2 (2010.0-2014.0) and ESA operational (2014.0-2019.0) OTL estimates: GPS (blue), GLONASS (orange), GPS+GLONASS (green) constellation modes present. Grey crosses are as per Figure 3.

processing. Considering the $S_2$, $\|\Delta Z_{res}\|$ shows the very same form of bias as previously seen with 2010.0-2014.0 dataset which suggests symmetric deviation of repro2 and operational products solutions from modelled value.

## 6 Conclusions

We expand the GPS-only methodology of ocean tide loading displacement estimation described in Penna et al. (2015) with GLONASS constellation and assess the performance of GPS and GLONASS for estimation of eight major ocean tide loading constituents in stand-alone modes and in a combined GPS+GLONASS mode. We examine data from 21 sites from the UK and western Europe over a period of 2010.0-2014.0 through processing data in kinematic PPP using products from three different analysis centres: CODE, ESA and JPL. The latter was also used to assess the effect of GPS ambiguity fixing on estimated

ocean tide loading displacement. All solutions were intercompared to gain an insight into the sensitivies of the constituent estimates to different choices of satellite orbit and clock products and constellation configurations.

We find that no single constellation mode solution for all eight major tidal constituents exists. However, we do find that GLONASS-based estimates show a comparable level of performance to ambiguity-free GPS for $M_2$, $N_2$, $O_1$, $P_1$ and $Q_1$ while showing improved results for $K_2$ and $K_1$. Alternatively, this optimal solution can be constructed from the combination of

constellation modes for each constituent and component for the case of GPS and GLONASS presence.

We show that ambiguity-free GPS+GLONASS solutions show a similar level of precision as GPS with ambiguities resolved (GPS AR), with $P_1$ estimates using GPS+GLONASS showing improved precision and stability. The $K_2$ and $K_1$ constituents, which are known to be problematic in GPS solutions, are still unusable in GPS+GLONASS solutions, due to propagation of GPS related errors. The $S_2$ constituent also cannot be reliably recovered with GPS+GLONASS as GLONASS shows de-

pendency between estimates and elevation cutoff angle. GPS-based estimates of $S_2$ show a constant bias when ambiguity resolution is not implemented, but this is removed by resolving the ambiguities to integers. GLONASS-based estimates show





a comparable level of performance to ambiguity-free GPS for $M_2$, $N_2$, $O_1$, $P_1$ and $Q_1$ while showing improved results for $K_2$ and $K_1$.

Additional comparison of OTL estimates from reprocessed and operational products shows that GLONASS estimates of $K_2$
and $K_1$ show differences in the up and, to the lesser extent in the east, components when using different products.

Considering the above, we suggest that estimation of $K_1$ and $K_2$ constituents is best undertaken using GLONASS only solutions with an emphasis towards north component where it is most stable. $M_2$, $S_2$, $N_2$, $O_1$ and $Q_1$ can be reliably estimated from combined GPS+GLONASS or GPS AR solutions while $P_1$ is best with GPS+GLONASS.

Integer ambiguity resolution was not possible in the GLONASS or GPS+GLONASS solutions tested here due to limitations
in the products available. However, evidence from our GPS AR testing suggests that further increases in precision and stability will be seen when AR fixing can be performed which should have a positive impact on solar-related constituents.

*Code and data availability.* GNSS data were obtained from the Natural Environment Research Council (NERC) British Isles continuous GNSS Facility (BIGF), www.bigf.ac.uk and the International GNSS Service (IGS), www.igs.org. OTL values and TPXO.7.2 OTL grid were obtained from free ocean tide loading provider, holt.oso.chalmers.se/loading/. CODE REPRO_2015 and CODE MGEX orbit and clock
products were obtained from the University of Bern, ftp.aiub.unibe.ch/REPRO_2015/ and ftp.aiub.unibe.ch/CODE_MGEX/ respectively, ESA repro2 and operational from the Crustal Dynamics Data Information System (CDDIS), JPL repro 2.1 and repro 3.0 from NASA Jet Propulsion Laboratory, sideshow.jpl.nasa.gov/pub/.

GipsyX binaries were provided under license from JPL. Eterna tidal analysis and prediction software with source code was acquired from International Geodynamics and Earth Tide Service (IGETS), igets.u-strasbg.fr/soft_and_tool.php. The source code of GipsyX wrapper
developed to facilitate the processing, analyse output and undertake plotting can be found at: github.com/bmatv/GipsyX_Wrapper.

*Author contributions.* BM facilitated processing of the GNSS data with GipsyX and analysis of the resulting timeseries under the supervision of MK and CW. All authors contributed to the discussion of the results and writing of the manuscript.

*Competing interests.* The authors declare that they have no conflicts of interest.

*Acknowledgements.* The services of the Natural Environment Research Council (NERC) British Isles continuous GNSS Facility (BIGF),
www.bigf.ac.uk, in providing archived GNSS data to this study, are gratefully acknowledged. We are grateful to NASA Jet Propulsion Laboratory for the GipsyX software, products and support. We thank IGS, www.igs.org, for providing GNSS data and ESA reprocessed products; CODE, www.aiub.unibe.ch, for providing reprocessed products. We thank Klaus Schueller for advice and discussion on Eterna software. The services of TPAC High Performance Computing Facility are acknowledged gratefully. We gratefully acknowledge Machiel





Bos whose support and discussion were vital for the project. We thank the Free Ocean Loading Provider for the OTL computation services

and for providing the OTL grid that was used in Figure 1.



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
