# Peer review of "Estimating ocean tide loading displacements with GPS and GLONASS"

_Solid Earth, 2020_

## Referee Comment (RC1) · Anonymous Referee #1 · 21 May 2020

This paper characterizes the impact of GLONASS, when supplementing GPS, on the recovery of ocean tidal loading (OTL) signals from geodetic stations. GPS techniques are somewhat impaired in estimating K2 and K1 tide constituents, due to errors that resonate with the sidereal repeat periods of the GPS satellite orbits and the constellation geometry. The results of this paper support that GLONASS, sampling for which is not sidereal, can significantly overcome this weakness. While the results are based on data from a European network, they are likely more widely applicable. The results hint at the potential benefits of combining observations from other GNSS systems (e.g., Galileo, Beidou, in addition to GLONASS—with their varying geometries and repeat periods—for addressing some of the most demanding geodetic problems. The outcome also provides new perspectives on potential scientific contributions of the

[Figure]

GLONASS system. The approach is thorough and densely documented, but seems not notably inventive, leaning heavily on the prescription of Penna et al. (2015). While the results are not unexpected, they are certainly noteworthy and warrant publication.

I am a bit puzzled, however, by certain aspects of the solution strategy. Why, for example, is the OTL signal removed a priori in the GipsyX PPP solutions (using FES2004_Gbe, cf. Line 106)? Later, the OTL signal is added back in (imperfectly) using HARDISP (Line 140). Why not simply turn off the OTL model in GipsyX and attempt to recover the full expression of OTL, thus removing any doubt that the a priori model is somehow favored via constraint? In this scenario, one could adopt a less constrained random walk (RW) for recovery of the position. Why indeed does the 3.2 mm/sqrt(s) recipe from Penna et al. (2015) again emerge as the optimal constraint for the RW position estimate? Figure 2 does not seem to suggest that the amplitudes of the recovered OTL constituents become unbounded with higher process noise. Indeed, they seem to stabilize. One wonders what the outcome would be with a more disruptive estimation strategy that allows the position to move more freely and independently (with no background OTL model). Perhaps I am missing something here, and would of course invite the authors to clarify.

I think the estimated clocks and zenith troposphere also provide clues to this sensitivity problem. One could probe the time series of these "nuisance" parameters for signs of energy at the sidereal periods linked to the GPS repeats.

One other suggestion is to consider relegating additional detailed discussion to the supplementary material. The salient points sometimes get obscured by the detailed descriptions of the results and cases. The paper is otherwise well written.

---

## Referee Comment (RC2) · Anonymous Referee #2 · 2 Jun 2020

This paper considers the estimation and quality control of ocean tide loading (OTL) displacements estimated using GPS, GLONASS, and GPS+GLONASS. The paper is largely incremental in nature, closely following the GPS OTL displacement measurement work of Penna et al (2015) and using the same regional (largely UK) data set, but extending their work to include more tidal constituents, three coordinate components, some different satellite orbit and clock products, and the GLONASS constellation and its combination with GPS. Such GLONASS and GPS+GLONASS OTL measurement demonstrations have yet to be published, so the dissemination of these tests is of interest and benefit to the scientific community. I do however feel that there are quite a lot of aspects (as detailed below) that need to be improved and/or carefully checked before the paper can be deemed acceptable for publication.

Main points (in sequential order):

1. Section 1 Introduction: It would be helpful to make clearer what the main aims and objectives of the paper are, in order to better put the many documented tests into context. The Introduction provides a generally useful overview of previous works and some limitations, but the last paragraph fails to build on this, just describing what will be included in the paper, without explaining why such tests are being done, or what the paper is seeking to achieve and why. Please also revise some of the description about why anelasticity etc has not been studied until recently, as it is not the case (L41) that limitations in PREM were the reason. The limitations were the accuracy of the ocean tide models (as recognised on L47) and the quality and availability of GNSS measurements/processing.

2. Section 2 Dataset selection: From Figure 1, one cannot see what M2 up OTL displacement the actual GNSS stations experience, except for those stations right on the coastline. This figure would be much more useful if the displacement over the land were also plotted. Note that the ocean tide model series is TPXO, so the model is TPXO7.2, not TPXO.7.2 (Figure 1 caption and L368). All stations in the figures and study are in Europe (L84), so please describe the stations' geographical distribution more accurately. Regarding the selected stations, given that the GNSS minus model residual is being used as a quality indicator (with zero the aim), some indication of expected errors based on previous studies should be included. Are some of the larger residuals later shown likely to be associated ocean tide model or Green's function errors and hence not just indicators of GNSS measurement quality? E.g. Bos et al (2015) state that they excluded the GNSS stations CARI and SWAS from their geophysical results because of large ocean tide model discrepancies in the Bristol Channel.

3. Section 3 GNSS data processing: Please clarify what exact OTL model was used in the GipsyX processing and what was used in the subsequent results. It seems strange in a technical paper such as this where the variables are the GNSS constellations, orbits etc, to use some form of OTL model remove-restore procedure. Was

hardisp used with FES2004 and Gutenberg-Bullen in the GipsyX processing? If so, I am not clear why it is then said on L140 that the "OTL displacements modelled in GipsyX were added back using HARDISP" when referring to the post-processed time series? So if FES2004/Gutenberg-Bullen was used in the GipsyX processing, the carrier phase residuals shown in Figure 2 presumably contain some FES2004/Gutenberg-Bullen errors, while the OTL up displacement also shown in Figure 2 is a residual to FES2014b/STW105d (how were these computed as, unlike for FES2004_GBe, there is no explicit statement of their source)? Hence are results from different OTL models displayed in the same figure, which would seem to be inconsistent? I do not follow the statement on L159-160 that FES2014b_STW105d is used unless stated otherwise, but then in Section 4, it is suggested that FES2004/Gutenberg-Bullen was used. Exactly what was used for which test needs to be clearly explained as it is hard to fully interpret the results at present, but why the need to mix and match? Please also note that CODE and ESA orbits provided via the IGS are in a centre of network (CN) frame, rather than CE. Regarding orbit and clock products, the authors should make clear that their statements regarding PPP AR and particular products are in the context of GipsyX, and not necessarily other GNSS processing softwares. On L136, please provide details as to what is meant by "The raw 4-yr timeseries were filtered".

4. Section 4 Process noise optimization: Figure 2 shows that the M2 residual varies with the coordinate process noise, which is to be expected, yet the synthetic signal introduced (what amplitude?) is recovered with apparent zero error with all process noise ranges. I think the authors need to carefully check their processing and implementation: there are various ways by which the effect of a synthetic signal can be introduced to GNSS processing, but intuitively if a very small coordinate process noise is applied, then one would not expect to be able to recover it. Then in the supplement, the authors show that they recover a 6 mm amplitude synthetic signal perfectly with a very small process noise (which is counter-intuitive), but not at all with a larger process noise. Note that Martens et al (2016), which the authors cite but not in relation to synthetic signals, obtain results corroborating those of Penna et al (2015), in that the synthetic

signal is not recovered if a very small process noise is applied. The authors also state in the supplement that in kinematic positioning constant nominal coordinates should be used, but this is not always appropriate, for example the case of a rapidly moving GNSS receiver.

5. Section 5 and Section 5.1: Overall, I found the order and logic of all these tests a bit piecemeal and not particularly well flowing and ordered. To improve the readability, I encourage the authors to improve the description of why each test was undertaken, and how they build on the preceding ones. Section 5.1, although entitled "Effect of using GLONASS", is really a generic introduction to the subsequent tests, as almost all of them assess the effects of GLONASS and GPS+GLONASS in some way. So I think this section would be better as an introductory Section 5, that then leads into the others. For example, it seems odd to include so little discussion in Section 5.1 on the benefits of using GLONASS for K2 and K1, and the overall performance of GLONASS and GPS+GLONASS. I was left wondering where this was given the section heading, until I saw a separate section called "K2 and K1 constituents". Please include quantification in the description of improvements, rather than simply saying, for example, "smaller". Any suggestions as to why there are variations in residuals among the different constellations and ambiguity fixing approach for the different coordinate components?

6. Section 5.3: It seems strange that the complete opposite of larger vs smaller GPS or GLONASS residuals arises when using CODE and ESA orbits and clocks. Any more explanation for this? For instance, I am not clear why the correction of atmospheric tides in the CODE products will improve GPS but not GLONASS?

7. Section 5.6 Noise and uncertainty: This paper concerns the generation and analysis of OTL displacement residuals. So if the modelled OTL displacements applied are in a frame compatible with the orbital products, as I assume has been done throughout the paper (this is implied on L106-107), then are the residuals (and their uncertainties) actually sensitive to the frame given the model and GNSS measurement precisions?

Figures 3-5, in which JPL and CODE/ESA residuals are compared, do not suggest the residuals themselves are. It is not clear how the JPL products in CM "provide a significant advantage" (L288) over ESA and CODE products in CE (CN) in relation to phase uncertainties. The authors state that the effect will depend on the constituent's amplitude, but the CE-CM difference (and hence whether the CM value or CE value has the larger amplitude) will depend on the location globally. What exactly has been applied in the creation of the two panes in Figure 8? I am also puzzled by the results for the averaging of the amplitude uncertainties across all constituents. It seems odd that the K1 and K2 amplitude uncertainties are commensurate in precision with the lunar constituents such as M2 and N2, given the errors and associated problems with K1 and K2.

8. Section 5.8 Timeseries length: The authors should summarise the status of the various IGS Analysis Centre reprocessing and operational products for the reader who might not have intricate knowledge of what was done and when, and hence what timespan such products cover. What are the key differences between the repro2 products of 2010.0-2014.0 and those used for 2014.0-2019.0? If the operational products adopted repro2 procedures for much of this time, then one would not expect noticeable changes, but if major changes arose in the operational products during the 5 year window, then comparing time series lengths involves multiple variables, when the desire is to just vary the time series length. To me it would be better to describe the test shown in Figure 12 first, in order to evaluate the impact of any differences in products, as well as showing another test of measurement noise. The authors say they found no significant differences at M2, N2, O1, P1 and Q1, but Figure 12 suggests there is a noticeable difference at P1 for GPS, perhaps unsurprisingly given P1 challenges discussed earlier in the paper. It is stated on L335-336 that GLONASS showed significant differences for K1 and K2, but Figure 12 suggests the differences are bigger for GPS, so what is implied here? It is stated on L325 that all eight constituents' variation with time series length was assessed, but then only S2 and K1 are shown (in Figure 11) and discussed. I had to search through the supplement to find the other constituents, but this needs to

be made clear in the paper itself, and state why the emphasis in the paper is on S2 and K1. Many of the constituents appear (from inspection of the supplement) stable over time, which suggests that even if there are changes in the products, they are not having an impact. I think more should be made of this point in this section, as it is a positive result. I am unclear what is meant by the sentence on L338-339 – please rewrite.

9. Supplement: It is not clear what new information related to tidal cusps has been found by the authors or why they have raised this point. Section 6.1.1 of the Penna et al (2015) study does not state that there are tidal cusps, but discusses that any slight gradual increases in power around M2 are likely caused by spectral leakage.

Other points:

Please provide a concise statement defining the bounds of the box and whisker plots used throughout the paper, as there can be different conventions used. E.g. on L191 I did not follow what is meant as the stated lower bound.

Please decide on "vector difference" or "vector distance", and use consistently throughout the paper.

L58: Not clear why ESA and CODE orbit/clock products are specifically mentioned here but those from other IGS Analysis Centres are not. The justification for these two is provided in Section 3, but on L58 the mentions are out of place and premature without more explanation.

L203: Please explain in what way different elevation cut-off angles will modulate the expression of signal multipath into solutions. I think you mean by this that you expect less multipath with larger cut-offs?

L209-210: What is meant by the sentence "For the. . . (including K1 and K2)"? It does not make sense, and what does increase the stability mean?

L211-213: Sentence contradicts itself. Please clarify.

L221-222: By "partial expression", do you mean partially propagate?

L253-255: Please refer to Figure 3 so that the reader can ascertain what is being referred to.

L256: Is 'tightest' a technical term?

L258-259: What is meant by "The GLONASS K1 east is not true"?

L270-271: What is meant by "better consistency between products"? I think it better to say that GLONASS gives smaller values, rather than being "preferred".

L313-314: Please state where (this paper?) the "similar behaviour previously observed with ESA products" was.

L315-316: Please refer to the figure in question, and quantify "completely" and "slight increase".

L316: It is mentioned that the implementation of ambiguity resolution results in a slight increase in the median of the up component. Any explanation for this? From inspection of Figure 10, I am not convinced anything of significance is showing compared with the float solutions.

The authors need to carefully go over the English and presentation to improve the readability of the paper (particular the use of "the") and ensure all acronyms and abbreviations are defined. Non-exclusive examples (in Sections 1-3 only) include:

L3: "close to several" -> "close to that of several"

L4: is -> are

L5: over -> of

L6: "Western Europe" -> "western Europe"

L17: hyphen not needed in "solid Earth"

L24: Incorrect reference format

L31 (and elsewhere): The reference should be Wang et al (2020) not 2019. L451-453 is out of date

L33: I would put "predominantly" before "estimated"

L49: "areas", rather than "conditions"?

L54: I would add "period" and "orbital"

L55: "constellation period" -> "constellation repeat period"

L62: "constellation period" -> "constellation repeat period"

L62: remove spaces before the "11" and "8"

L79: "observation" -> "observations"

L80: insert "the" before "selected"

L105: change to "within each processing"

L107: No need to describe the file format

L108: change to "using the free" L110: change "single products solution" to "single product's solution"

L113: change "Products" to "products"

L134: no need to describe the basic merging / concatenation of IGS files

L143: insert "the" before "processing"

L146: insert "the" before "Eterna"

L148: remove "the" before "solid Earth"

L151: remove "using the procedure"

L158: first bracket is in the wrong place

---

## Author Comment (AC1) · 13 Jul 2020

**Response to Reviewer 1's remarks on "Estimating ocean tide loading displacements with GPS and GLONASS" se-2020-22**

We thank the reviewer and editor for constructive remarks which we respond to below in **bold.**

**NOTE: On additional check we found some minor errors in our plotting routine which slightly affected figures 3, 6, 9, 10, 11 and their supplementary versions. We updated the figures and their analysis. These changes do not affect the conclusions.**

**The key updates are:**

- **Fig 3: GPS AR (JPL) the variance in up is smaller by ~1.3 mm for S2 and 1 mm for K1 but larger by ~2.5 mm for K2**
- **Fig 6. CODE GPS a ~1 mm bias in S2 appears for solutions with cutoff angle >= 10 while median values change marginally.**
- **Fig 9. GPS (JPL) a ~0.7 mm bias in S2 east appears. The variances of K1 and K2 become smaller by ~2 mm**
- **Fig 10. GPS AR (JPL) shows a clearer linear dependency with cutoff angle.**
- **Fig 11. A constant ~0.3 mm S2 east bias appears**

Why, for example, is the OTL signal removed a priori in the GipsyX PPP solutions (using FES2004_Gbe, cf. Line 106)? Later, the OTL signal is added back in (imperfectly) using HARDISP (Line 140). Why not simply turn off the OTL model in GipsyX and attempt to recover the full expression of OTL, thus removing any doubt that the a priori model is somehow favoured via constraint? In this scenario, one could adopt a less constrained random walk (RW) for recovery of the position. Why indeed does the 3.2 mm/sqrt(s) recipe from Penna et al. (2015) again emerge as the optimal constraint for the RW position estimate? Figure 2 does not seem to suggest that the amplitudes of the recovered OTL constituents become unbounded with higher process noise. Indeed, they seem to stabilize. One wonders what the outcome would be with a more disruptive estimation strategy that allows the position to move more freely and independently (with no background OTL model). Perhaps I am missing something here, and would of course invite the authors to clarify.

**The approach we use is the same as adopted by Penna et al (2015). This approach is adopted to 1) remove in a convenient way the companion tides using the hardisp routine and 2) it reduces time series variability and hence allows for the selection of a tighter time series process noise which helps reduce noise.**

**Our finding of the optimal process noise setting is the same as Penna et al (2015). This is not surprising to us given that we use a similar site which is therefore subject to very similar site noise, tropospheric variations and OTL. The key difference is the time period and the addition of GLONASS data. That the findings on the optimal parameters are the same as Penna et al suggests that data noise or product noise is less important than the geophysical signal. Note that we choose a slightly conservative process noise that allows for sites to have significant freedom. On additional testing, we found that the overall vector difference for M2 between the approach used in Penna et al. (2015) and approach with OTL not modelled during processing run is ~0.1 mm (Fig. R1), however, this difference also includes phase variations due to companion tides and hence this is an upper bound on the difference. Given this is a standard and widely used approach we do not modify the manuscript.**

[Figure]

**Figure R1. The effect of varying coordinate process noise at test site CAMO for the up component (2010.0 – 2014.0), performed with ESA repro2 products with OTL modelling enabled (left), disabled (centre and right) during PPP run. The right plot ZWDstatic was computed relative a static solution with OTL modelling enabled.**

I think the estimated clocks and zenith troposphere also provide clues to this sensitivity problem. One could probe the time series of these "nuisance" parameters for signs of energy at the sidereal periods linked to the GPS repeats.

**We tested the difference in estimated wet zenith delay between solutions with and without modelled OTL and found the difference to be negligible thus demonstrating that the zenith delay is not absorbing OTL with the chosen settings. The tropospheric gradients were also analysed, and the differences were again negligible. Given these and earlier tests by us and Penna et al. (2015) we are content that the solutions are optimal. To test if any tidal signal was absorbed by the wet zenith or gradient parameters, we repeated the solutions but without OTL modelled at the observation level and compared the sets of parameters. We found negligible differences suggesting that OTL is not being absorbed into these parameters without chosen process noise settings.**

One other suggestion is to consider relegating additional detailed discussion to the supplementary material. The salient points sometimes get obscured by the detailed descriptions of the results and cases. The paper is otherwise well written.

**We have reviewed the manuscript for overly detailed discussion of the results and editing the text appropriately throughout.**

---

## Author Comment (AC2) · 13 Jul 2020

**Response to Reviewer 2's remarks on "Estimating ocean tide loading displacements with GPS and GLONASS" se-2020-22**

We thank the reviewer for meticulous review and constructive remarks that greatly impacted the quality of the prepared publication. We break down the points of the reviewer into sub blocks and respond in **bold**.

**NOTE: On additional check we found some minor errors in our plotting routine which slightly affected figures 3, 6, 9, 10, 11 and their supplementary versions. We updated the figures and their analysis. These changes do not affect the conclusions.**

**The key updates are:**

- **Fig 3: GPS AR (JPL) the variance in up is smaller by ~1.3 mm for S2 and 1 mm for K1 but larger by ~2.5 mm for K2**
- **Fig 6. CODE GPS a ~1 mm bias in S2 appears for solutions with cutoff angle >= 10 while median values change marginally.**
- **Fig 9. GPS (JPL) a ~0.7 mm bias in S2 east appears. The variances of K1 and K2 become smaller by ~2 mm**
- **Fig 10. GPS AR (JPL) shows a clearer linear dependency with cutoff angle.**
- **Fig 11. A constant ~0.3 mm S2 east bias appears**

1. Section 1 Introduction:
    1.1. It would be helpful to make clearer what the main aims and objectives of the paper are, in order to better put the many documented tests into context. The Introduction provides a generally useful overview of previous works and some limitations, but the last paragraph fails to build on this, just describing what will be included in the paper, without explaining why such tests are being done, or what the paper is seeking to achieve and why.
    **The last paragraph of the Introduction was rewritten to clearly describe the tests and objectives. "We seek to improve estimates of OTL displacement from continuous GNSS data. especially for constituents that are subject to systematic error in GPS-only solutions (e.g., K1, K2, S2, P1) as found in previous studies** (Allinson, 2004; King, 2006; Yuan & Chao, 2012)**. We do this by using both GLONASS and GPS data to estimate amplitudes and phases for the eight major OTL constituents (M2,S2,N2,K2,K1,O1,P1,Q1). Our work focuses particularly on understanding the sensitivity of estimates to different processing choices."**

    1.2. Please also revise some of the description about why anelasticity etc has not been studied until recently, as it is not the case (L41) that limitations in PREM were the reason. The limitations were the accuracy of the ocean tide models (as recognised on L47) and the quality and availability of GNSS measurements/processing.
    **We revised this block to correct the limitations issue and better reflect the logical order of advancements. "This type of investigation has not been easily done previously due to various limiting factors such as the accuracy of ocean tide models and the quality and availability of GNSS observations. Accuracy advances achieved with recent altimeter-constrained ocean-tide models (Stammer et al., 2014) and evolving GNSS networks has enabled studies to identify limitations in the global seismic Preliminary Reference Earth Model (PREM) ..."**
    **The following paragraph was revised and appended to make this section more concise.**

2. Section 2 Dataset selection:
    2.1. From Figure 1, one cannot see what M2 up OTL displacement the actual GNSS stations experience, except for those stations right on the coastline. This figure would be much more useful if the displacement over the land were also plotted.
    **We updated the design of the Figure 1 with over-land OTL displacement.**

    2.2. Note that the ocean tide model series is TPXO, so the model is TPXO7.2, not TPXO.7.2 (Figure 1 caption and L368).
    **Corrected. Originally, we used naming as in free OTL provider.**

    2.3. All stations in the figures and study are in Europe (L84), so please describe the stations' geographical distribution more accurately.
    **The dataset description was updated to describe the locations: Of the 21 stations, 14 stations are in south-west England: covering both sides of Bristol channel (ANLX, SWAS, CARI, CAMO, PADT, APPL, TAUT) and northern coast of English Channel up to Herstmonceux (PMTH, PRAE, EXMO, PBIL, POOL, CHIO, SANO, HERT) with one site (BRST) in the south. Two sites are in northern England (WEAR, LOFT), two in Scotland (LERI, BRAE) with one site in central Europe (ZIM2). All sites are equipped with GPS+GLONASS receivers. Note that sites CAMO, LERI and ZIM2 sites replace CAMB, LERW and ZIMM respectively, that were used by Penna et al. (2015) because the earlier sites were replaced with newer ones, including adding GLONASS tracking.**

    2.4. Regarding the selected stations, given that the GNSS minus model residual is being used as a quality indicator (with zero the aim), some indication of expected errors based on previous studies should be included.
    **We have added a sentence which summarises the typical magnitude the differences between observations and models (~0.5-2mm depending on constituent and location) based on previous studies (e.g., Yuan et al., 2013).**

    2.5. Are some of the larger residuals later shown likely to be associated ocean tide model or Green's function errors and hence not just indicators of GNSS measurement quality? E.g. Bos et al (2015) state that they excluded the GNSS stations CARI and SWAS from their geophysical results because of large ocean tide model discrepancies in the Bristol Channel.
    **Here we assess the difference between various solutions using different products and constellation configurations , in terms of the effect on the residual OTL computed relative to some fixed modelled OTL. This means that the biases due to ocean tide**

**model or Green's functions are differenced. Thus, CARI and SWAS were not excluded.**

3. Section 3 GNSS data processing:
   Please clarify what exact OTL model was used in the GipsyX processing and what was used in the subsequent results. It seems strange in a technical paper such as this where the variables are the GNSS constellations, orbits etc, to use some form of OTL model remove-restore procedure. Was hardisp used with FES2004 and Gutenberg-Bullen in the GipsyX processing? If so, I am not clear why it is then said on L140 that the "OTL displacements modelled in GipsyX were added back using HARDISP" when referring to the post-processed time series?
   **We follow the procedure of Penna et al. (2015) here who in turn follow the work of King et al. (2000, 2003).As summarised in the response to reviewer #1, when we tested the alternative approach (e.g., comparison of techniques used by Penna et al. (2015) with those used by Martens et al. (2016)) we found differences of no more than 0.1mm (and this includes the effect of removing the companion tides so is a an upper bound). . We have amended the text at the first mention of FES2004 and GB to clarify that we later restore this signal.**

   3.1. So if FES2004/Gutenberg-Bullen was used in the GipsyX processing, the carrier phase residuals shown in Figure 2 presumably contain some FES2004/Gutenberg-Bullen errors,
   **This is not correct given we use a kinematic processing approach and hence mismodelled OTL is simply reflected in the time series. If the coordinate solution was 24hr rather than 5min, these data would affect the residuals.**

   3.2. while the OTL up displacement also shown in Figure 2 is a residual to FES2014b/STW105d (how were these computed as, unlike for FES2004_GBe, there is no explicit statement of their source)?
   **L110 updated to: All OTL values used in the publication were generated using CARGA software at the free ocean tide loading provider (http://holt.oso.chalmers.se/loading) (Bos & Baker, 2005).**

   3.3. Hence are results from different OTL models displayed in the same figure, which would seem to be inconsistent?
   **The residual displacement shown in Fig 2 is relative to FES2004_Gbe. This is mentioned on L173. We added "$\|Z_{res}\|$ is relative to FES2004_Gbe." to the figure caption.**

   3.4. I do not follow the statement on L159-160 that FES2014b_STW105d is used unless stated otherwise, but then in Section 4, it is suggested that FES2004/Gutenberg-Bullen was used. Exactly what was used for which test needs to be clearly explained as it is hard to fully interpret the results at present, but why the need to mix and match?
   **The M2 residual OTL values used in the test could be computed through analysis of raw timeseries. We, however, restore the OTL so need to subtract it to get the residual. If not restored, the phase STD values may be slightly distorted. We removed "computed by differencing observed OTL with FES2004_Gbe theoretical values ("M2")" to prevent confusion.**
   **L153 was updated to: "We then computed the vector difference between the *reconstructed observed* OTL and that predicted." to highlight the reconstructed full OTL signal used in further analysis.**

   3.5. Please also note that CODE and ESA orbits provided via the IGS are in a centre of network (CN) frame, rather than CE. Regarding orbit and clock products, the authors should make clear that their statements regarding PPP AR and particular products are in the context of GipsyX, and not necessarily other GNSS processing softwares.
   **While IGS orbits are in CN on a daily basis orbit products at sub-daily timescales are naturally in CM unless they are otherwise corrected for the CE-CM motion. The frame for sub-daily OTL is not dependent on a network of stations. CN is therefore not relevant to our study. We have added that "The findings in our paper are in the context of GipsyX software and solutions derived using other software may produce different results especially if the underlying model choices differ".**

   3.6. On L136, please provide details as to what is meant by "The raw 4-yr timeseries were filtered".
   **Changed to: Outliers were filtered from the raw 4-yr timeseries using two consecutive outlier-detection strategies: rejecting epochs with extreme clock bias values (>3x10³ m) or where the XYZ $\sigma$ was over 0.1 m; and then rejecting epochs with residuals to a linear trend larger than three standard deviations per coordinate component.**

4. Section 4 Process noise optimization:
   4.1. Figure 2 shows that the M2 residual varies with the coordinate process noise, which is to be expected, yet the synthetic signal introduced (what amplitude?) is recovered with apparent zero error with all process noise ranges.
   **see response to issue 4.2 and 4.3. Please note that the differences between our work and those of previous work are immaterial to the results and hence why this discussion is in the supplementary material.**

   4.2. I think the authors need to carefully check their processing and implementation: there are various ways by which the effect of a synthetic signal can be introduced to GNSS processing, but intuitively if a very small coordinate process noise is applied, then one would not expect to be able to recover it.
   **We clarify here that the synthetic signal is introduced as a nominal or a priori signal in GipsyX. As such, solutions with zero process noise, to take an extreme, are required to exactly follow the a priori since in this case no adjustment of the prior is allowed. That is, the solutions cannot deviate from the nominal. In that case, we regard the error should be zero (synth_err=0). The same situation effectively applies for very small (tight) process noise. We have now clarified in the text that this is the approach in GipsyX. We explain further under 4.3 the reason for the difference with previous studies.**

4.3. Then in the supplement, the authors show that they recover a 6 mm amplitude synthetic signal perfectly with a very small process noise (which is counter-intuitive), but not at all with a larger process noise. Note that Martens et al (2016), which the authors cite but not in relation to synthetic signals, obtain results corroborating those of Penna et al (2015), in that the synthetic signal is not recovered if a very small process noise is applied.

**We think this difference comes from either the use of an unintuitive (to us) definition of "error" in previous studies or from a misunderstanding in those studies of the output of a GIPSY routine. Our logic of how GipsyX works is outlined in 4.2 above. The opposite extreme is to have very high process noise in which case the solution adjusts away from the nominal to the solution governed by the data. In that case, the difference between the nominal and the solution should reflect the signal introduced in the nominal. We outline the problem in previous studies next: Let's take the smallest coordinate processing noise value on Fig 3 in Penna et al. (2015) which is 3.2 mm/sqrt(s). The synth_err parameter in their Figure 3 equals 6 mm, which essentially means that no synthetic signal was recovered given the synthetic signal was 6mm. If we repeat this test and then convert the XYZ coordinate solutions from the GIPSY output data file (tdptable) to ENU using the GIPSY xyz2env/xyz2llh script, (tdp2envDiff.py GipsyX) we find a value of synth_err of 0 mm not 6mm. However, if we plot the XYZ coordinate components from the output tdptable directly, they show the clear synthetic signal that we introduced. How is this possible? The answer lies in the xyz2env script which differences XYZ values (column 2 of the tdpfile) with their nominals (column 1 of the tdpfile) before rotation. We know that our timeseries with that tight noise will perfectly resemble the nominals we introduced (synth signal), so if we difference it with the nominals we get a straight line – cancelling out the signal and producing an apparent time series that is a straight line and hence, when differenced from the original signal, produces a 6 mm synth_err . We resolved this by creating a custom xyz2env script and we get the expected results – inverted dependency. As we note above, the points in 4.1, 4.2, 4.3 have no direct bearing on the conclusions of our work or previous work.**

**added "We only present results without a synthetic signal introduced into the sites' nominal location, with further discussion in the supplementary material (Fig. S6)."**
**removed "The only difference in our results to those of Penna et al. (2015) were for the "synth err" test, where our results are inverted (but without changing the magnitude); the reason for this is discussed in detail in the supplementary material."**

4.4. The authors also state in the supplement that in kinematic positioning constant nominal coordinates should be used, but this is not always appropriate, for example the case of a rapidly moving GNSS receiver.

**We do not consider rapidly moving GNSS receivers and cannot conceive of a case where OTL studies would use such a receiver. As such we do not make any changes to the text.**

5. Section 5 and Section 5.1:

5.1. Overall, I found the order and logic of all these tests a bit piecemeal and not particularly well flowing and ordered. To improve the readability, I encourage the authors to improve the description of why each test was undertaken, and how they build on the preceding ones.

**We added to the end of introductory section (5): We now explore the sensitivity of our solutions to different products and analysis choices starting with elevation cutoff angle sensitivity, directly related to the amount of possible multipath presence. We follow with intercomparison of various products solutions and assessing impact from integer ambiguity resolution (GPS only) to understand the possible associated inconsistencies. Finally, we test fidelity of Eterna software to various constituents and timeseries length.**
**We have also added a single sentence to the beginning of each test to explain how each test flows from the previous.**

5.2. Section 5.1, although entitled "Effect of using GLONASS", is really a generic introduction to the subsequent tests, as almost all of them assess the effects of GLONASS and GPS+GLONASS in some way. So I think this section would be better as an introductory Section 5, that then leads into the others. For example, it seems odd to include so little discussion in Section 5.1 on the benefits of using GLONASS for K2 and K1, and the overall performance of GLONASS and GPS+GLONASS. I was left wondering where this was given the section heading, until I saw a separate section called "K2 and K1 constituents".

**We updated 5.1 to a more general 5**

5.3. Please include quantification in the description of improvements, rather than simply saying, for example, "smaller". Any suggestions as to why there are variations in residuals among the different constellations and ambiguity fixing approach for the different coordinate components?

**The paragraph was revised to reflect the updated Fig. 6 and to quantify the estimates. While we want to only highlight the differences within multiple parameters that have great effect on OTL estimates, several factors such as satellite's orbit models and different handling of inter-frequency & inter-system biases (GLONASS) are of higher importance. Also, the period analysed here is the worst in terms of GLONASS availability as the constellation was only fully restored (24 SVs) in March 2010. The note on constellation restoration information was added.**

6. Section 5.3: It seems strange that the complete opposite of larger vs smaller GPS or GLONASS residuals arises when using CODE and ESA orbits and clocks. Any more explanation for this? For instance, I am not clear why the correction of atmospheric tides in the CODE products will improve GPS but not GLONASS?

**We suggest it can only come from satellite orbit modelling differences propagating to orbit and clocks errors and/or multipath, although the exact source is not clear to us and would require a substantial study focused on product generation. We have removed mention of atmospheric tides as, we agree, it cannot affect GPS and GLONASS differently.**

7. Section 5.6 Noise and uncertainty:

7.1. This paper concerns the generation and analysis of OTL displacement residuals. So if the modelled OTL displacements applied are in a frame compatible with the orbital products, as I assume has been done throughout the paper (this is implied on L106-107), then are the residuals (and their uncertainties) actually sensitive to the frame given the model and GNSS measurement precisions? Figures 3-5, in which JPL and CODE/ESA residuals are compared, do not suggest the residuals themselves are.

**Our assumption is that inconsistencies between frames are far too small to be noticeable. Additional testing (purely synthetic) suggests the possible presence of minor inconsistencies between CoM coefficients computed with FES2004 and FES2014b ocean tide models. However, they are concentrated within K1 and O1 constituents with magnitudes of ~0.2 mm in each component. This suggests that frame inconsistencies should be considered in the future, especially with MultiGNSS AR processing by correcting for residual CoM in CE solution due to limitations of FES2004 used according to IERS recommendations. This is a subtle detail that is not core to the paper and so we do not further extend the paper by adding a discussion.**

7.2. It is not clear how the JPL products in CM "provide a significant advantage" (L288) over ESA and CODE products in **CE (CN)** in relation to phase uncertainties. The authors state that the effect will depend on the constituent's amplitude, but the CE-CM difference (and hence whether the CM value or CE value has the larger amplitude) will depend on the location globally. What exactly has been applied in the creation of the two panes in Figure 8?

**Our remarks pertain to the study region. We took the standard deviation values of phase measurements returned by Eterna and averaged them per component and constituent. We added a short note to Fig. 8 caption.**

7.3. I am also puzzled by the results for the averaging of the amplitude uncertainties across all constituents. It seems odd that the K1 and K2 amplitude uncertainties are commensurate in precision with the lunar constituents such as M2 and N2, given the errors and associated problems with K1 and K2.

**The results were checked and confirmed suggesting that Eterna is not facing any limits related to the constituent.**

8. Section 5.8 Timeseries length:

8.1. The authors should summarise the status of the various IGS Analysis Centre reprocessing and operational products for the reader who might not have intricate knowledge of what was done and when, and hence what timespan such products cover. What are the key differences between the repro2 products of 2010.0-2014.0 and those used for 2014.0-2019.0?

**The goal of reprocessing campaign was to preserve consistency with operational products** (Griffiths, 2019)**, thus we assume the differences to be insignificant and not impacting the estimates. Text updated.**

8.2. If the operational products adopted repro2 procedures for much of this time, then one would not expect noticeable changes, but if major changes arose in the operational products during the 5 year window, then comparing time series lengths involves multiple variables, when the desire is to just vary the time series length. To me it would be better to describe the test shown in Figure 12 first, in order to evaluate the impact of any differences in products, as well as showing another test of measurement noise.

**We swapped figures 11 and 12 according to your recommendation.**

8.3. The authors say they found no significant differences at M2, N2, O1, P1 and Q1, but Figure 12 suggests there is a noticeable difference at P1 for GPS, perhaps unsurprisingly given P1 challenges discussed earlier in the paper.

**Based on previous results, we assumed that changing satellite orbit and clock products may produce substantial differences in problematic solar-related constituents (S2, K2, K1, P1). We better describe the repro/operational products test.**

8.4. It is stated on L335-336 that GLONASS showed significant differences for K1 and K2, but Figure 12 suggests the differences are bigger for GPS, so what is implied here? It is stated on L325 that all eight constituents' variation with time series length was assessed, but then only S2 and K1 are shown (in Figure 11) and discussed. I had to search through the supplement to find the other constituents, but this needs to be made clear in the paper itself, and state why the emphasis in the paper is on S2 and K1.

**The text has been updated to make the implication clearer that focus on S2 up is due to the evolution of inverted $\|Z_{res}\|$ bias and K1 up, as most problematic diurnal constituents.**

8.5. Many of the constituents appear (from inspection of the supplement) stable over time, which suggests that even if there are changes in the products, they are not having an impact. I think more should be made of this point in this section, as it is a positive result.

**We added this point. If constituents selected according to optimum constellation strategy, $\|Z_{res}\|$ appear (see Fig. S4) stable over time, which suggests that even if there are changes in the products, they are not having an impact with this methodology.**

8.6. I am unclear what is meant by the sentence on L338-339 – please rewrite.

**We completely reviewed this section to improve the description of the products.**

9. Supplement: It is not clear what new information related to tidal cusps has been found by the authors or why they have raised this point. Section 6.1.1 of the Penna et al (2015) study does not state that there are tidal cusps, but discusses that any slight gradual increases in power around M2 are likely caused by spectral leakage.

**We removed the sentence which mentioned tidal cusps as it is indeed irrelevant.**

10. Other points:

10.1. Please provide a concise statement defining the bounds of the box and whisker plots used throughout the paper, as there can be different conventions used. E.g. on L191 I did not follow what is meant as the stated lower bound.
**We remove "(25th percentile - 1.5\*interquartile range), which is present at all sites no matter how far inland."**
**Added at L159: "We utilize box-and-whisker plots to demonstrate the distribution of the estimates with the boxes defined by the interquartile range (IQR), with median as a horizontal line, and whiskers by an additional +- 1.5 \* IRQ.**

10.2. Please decide on "vector difference" or "vector distance", and use consistently throughout the paper.
**All instances of "vector distance" were changed to "vector difference"**

10.3. L58: Not clear why ESA and CODE orbit/clock products are specifically mentioned here but those from other IGS Analysis Centres are not. The justification for these two is provided in Section 3, but on L58 the mentions are out of place and premature without more explanation.
**This early mention of ESA and CODE products "*(ESA, but not CODE)*" was removed so not to confuse the reader.**

10.4. L203: Please explain in what way different elevation cut-off angles will modulate the expression of signal multipath into solutions. I think you mean by this that you expect less multipath with larger cut-offs?
**we updated L203: "Different elevation cutoffs will significantly alter the observation geometry as well as modulate the expression of signal multipath into solutions, decreasing its amount with higher cutoff value."**

10.5. L209-210: What is meant by the sentence "For the. . . (including K1 and K2)"? It does not make sense, and what does increase the stability mean?
**Stability means closeness of estimates between solutions (e.g. different cutoff angles). Part of the paragraph was updated to eliminate confusion and explicitly highlight the stability term.**
**We updated the sentence to "GPS+GLONASS shows smallest $\|\Delta Z_{res}\|$ between 7° and 20° estimates for S2 and P1 (0.31, 0.23 mm, respectively) and an additional decrease in $\|\Delta Z_{res}\|$ for M2, S2, N2, O1, Q1 in the up component, which indicate higher closeness of compared OTL values – higher stability with changing cutoff angle."**

10.6. L211-213: Sentence contradicts itself. Please clarify.
**L211-L213 updated to: The same comparison done for GPS AR (7° and 20° cutoff, JPL native products) shows largely improved stability in comparison to all GPS only ambiguity free solutions (Figure 4, bottom).**

10.7. L221-222: By "partial expression", do you mean partially propagate?
**Yes. Changed "...all satellite-related systematic errors will partially propagate into station-specific parameters"**

10.8. L253-255: Please refer to Figure 3 so that the reader can ascertain what is being referred to.
**Updated to : "As seen from Fig. 3, ...". In addition, removed $\Delta$ from $\|\Delta Z_{res}\|$, as Fig. 3 shows $\|Z_{res}\|$.**

10.9. L256: Is 'tightest' a technical term?
**term tightest was changed to "*have lowest variance*". Text was updated accordingly.**

10.10. L258-259: What is meant by "The GLONASS K1 east is not true"?
**The paragraph was updated**

10.11. L270-271: What is meant by "better consistency between products"? I think it better to say that GLONASS gives smaller values, rather than being "preferred".
**As suggested by the reviewer, we reviewed the sentence: "GLONASS returns smaller $\|Z_{res}\|$ in the K2 and K1 north and up components while east component might show better results with GPS+GLONASS (K1, CODE) but, due to higher closeness between products, GLONASS constellation is still preferred."**
**We leave preferred in the end as this applies to the optimum constellation configuration.**

10.12. L313-314: Please state where (this paper?) the "similar behaviour previously observed with ESA products" was.
**Added (Fig. 6)**

10.13. L315-316: Please refer to the figure in question, and quantify "completely" and "slight increase". L316: It is mentioned that the implementation of ambiguity resolution results in a slight increase in the median of the up component. Any explanation for this? From inspection of Figure 10, I am not convinced anything of significance is showing compared with the float solutions.
**We updated the paragraph: "Enabling integer ambiguity resolution (GPS AR) removes the ~1 mm S2 $\|Z_{res}\|$ bias completely at 7° and 10° elevation cutoff angles while leaving ~0.4 mm bias at 15° and 20° in the up component. Consequently, up $\|Z_{res}\|$ medians decrease by 1-2 mm depending on elevation cutoff angle. Based on this observation, we expect that utilising ambiguity resolution within PPP might help in solving, or at least minimising, the S2 $\|Z_{res}\|$ present in ESA GPS and CODE GLONASS solutions. Eliminating biases in GPS and GLONASS separately should increase the stability and consistency of GPS+GLONASS S2 $\|Z_{res}\|$."**

The authors need to carefully go over the English and presentation to improve the readability of the paper (particular the use of "the") and ensure all acronyms and abbreviations are defined. Non-exclusive examples (in Sections 1-3 only) include:
L3: "close to several" -> "close to that of several"        *Corrected*
L4: is -> are        *Corrected*
L5: over -> of        *Corrected*
L6: "Western Europe" -> "western Europe"        *Corrected*
L17: hyphen not needed in "solid Earth"        *Corrected*
L24: Incorrect reference format        *Corrected*
L31 (and elsewhere): The reference should be Wang et al (2020) not 2019. L451-453 is out of date        *Corrected*
L33: I would put "predominantly" before "estimated"        *Corrected*

L49: "areas", rather than "conditions"? **We might stick with conditions as in Stammer**

L54: I would add "period" and "orbital" **Not clear here**

L55: "constellation period" -> "constellation repeat period"     *Corrected*

L62: "constellation period" -> "constellation repeat period"     *Corrected*

L62: remove spaces before the "11" and "8"    *Corrected* **with tilde changed to $\sim$. Same with L79**

L79: "observation" -> "observations"     *Corrected*

L80: insert "the" before "selected"    *Corrected*

L105: change to "within each processing"    *Corrected*

L107: No need to describe the file format**. simplified to "The OTL values were generated using CARGA software at the free ocean tide loading provider (http://holt.oso.chalmers.se/loading) (Bos and Baker, 2005)."**

L108: change to "using the free"    *Corrected (see above)*

L110: change "single products solution" to "single product's solution"    *Corrected*

L113: change "Products" to "products"    *Corrected*

L134: no need to describe the basic merging / concatenation of IGS files    *Corrected:* **Removed "…and merging the IGS-standard 24-h orbits/clocks into 30-h where necessary"**

L143: insert "the" before "processing"    *Corrected*

L146: insert "the" before "Eterna"    *Corrected (check with Matt)*

L148: remove "the" before "solid Earth"    *Corrected*

L151: remove "using the procedure" *Corrected*

L158: first bracket is in the wrong place    *Corrected*

*We additionally reviewed the manuscript from section 4 onwards for minor issues as suggested by the reviewer*

---

## Author Response (AR2)

**Response to the Editor's remark on "Estimating ocean tide loading displacements with GPS and GLONASS" se-2020-22**

We thank the editor for the review and the brought-up issue:

*Both reviewers brought up questions regarding the remove-restore approach for the OTL signal. If I understand your reply to Reviewer-1's first point, you made some independent tests, using and not using the remove-restore, and you found differences of only 0.1 mm. Yet the paper doesn't mention this important point. So I recommend adding a sentence or two, somewhere around Line 150 or so, that addresses this issue, and confirms your results are not dependent on the prior FES2004 model.*

**We agree that the provided tests are important to connect the studies with and without the "remove-restore" procedure. We included this remark into the text of the publication (L149-152) as per the editor's recommendation.**

[revised manuscript text omitted]